# The Value of Out-of-distribution Data

## Abstract

More data is expected to help us generalize to a task. But real datasets can contain out-of-distribution (OOD) data; this can come in the form of heterogeneity such as intra-class variability but also in the form of temporal shifts or concept drifts. We demonstrate a counter-intuitive phenomenon for such problems: generalization error of the task can be a non-monotonic function of the number of OOD samples; a small number of OOD samples can improve generalization but if the number of OOD samples is beyond a threshold, then the generalization error can deteriorate. We also show that if we know which samples are OOD, then using a weighted objective between the target and OOD samples ensures that the generalization error decreases monotonically. We demonstrate and analyze this phenomenon using linear classifiers on synthetic datasets and medium-sized neural networks on vision benchmarks such as MNIST, CIFAR-10, CINIC-10, PACS, and DomainNet, and observe the effect data augmentation, hyperparameter optimization, and pre-training have on this behavior.

## 1 Introduction

We procure more data to improve generalization. The central assumption behind doing so—that we have baked into learning theory (Vapnik, 1998)—is that this data comes from the desired task. But this may not always be the case. Real data is often heterogeneous (Quinonero-Candela et al., 2008), this heterogeneity can arise from nuisances which are variables that do not inform the task at hand (say classification), e.g., geometric nuisances such as viewpoint, or semantic ones such as chairs of different shapes. Datasets curated at the Internet-scale Srivastava et al. (2022) may also be susceptible to erroneous annotations (resulting in label noise) (Frénay & Verleysen, 2013) or data poisoning attacks (Steinhardt et al., 2017). Such "out-of-distribution" (OOD) data, i.e., data that does not come from our desired task, can be detrimental to the performance of the learned model. In this work, we aim to study how OOD samples within datasets impact the generalization error on our desired task. Our contributions are as follows.

We demonstrate a counter-intuitive phenomenon: **generalization error on the target task is non-monotonic in the number of OOD samples.** In other words, there exist situations when a small number of OOD samples can improve the generalization error but if the number of OOD samples is beyond a threshold, then the generalization error deteriorates. This phenomenon is counter intuitive because one would expect the generalization error of the target task to deteriorate or improve monotonically upon the introduction of OOD samples. Our investigation shows that the threshold is different for different tasks and different neural architectures. In Remark 2, we provide an intuitive explanation for this non-monotonic behavior using the bias-variance trade-off.

**We present empirical evidence for the presence of non-monotonic trends in target generalization error in many popular datasets, ranging from MNIST, CIFAR-10, to PACS and DomainNet**. OOD samples within a curated dataset could lead to worse generalization error on the task for which the dataset was curated. We show that when OOD samples in the dataset are unknown, using strategies such as data-augmentation, hyperparameter optimization and pre-training, are not effective in eliminating the adverse impact of OOD data.

**We develop an algorithmic procedure to train on the target task that is resilient to OOD data.** If we know which samples within the dataset are OOD, e.g., using a two-sample test to check for changes in the distribution (Gretton et al., 2012), then we could mitigate the non-monotonic nature of the generalization error by ignoring the OOD samples. We show how one can do better: using

a weighted objective between the target and OOD samples, we can ensure that the generalization error on the target task decreases monotonically with the number of OOD samples. We empirically demonstrate the utility of this weighted objective on a variety of problems.

## 2 GENERALIZATION ERROR IS NON-MONOTONIC IN THE NUMBER OF OOD SAMPLES

We define a task $P$ as a joint distribution over the input domain $X$ and the output domain $Y$. We model the heterogeneity in the dataset as two distributions: $n$ samples drawn from a target task $P_t$ and $m$ samples drawn from an out-of-distribution (OOD) task $P_o$. We would like to minimize the generalization error $e_t(h) = \mathbb{E}_{(x,y) \sim P_t} [h(x) \neq y]$ on the target task. In order to do so, we may find a hypothesis that minimizes the empirical loss

$$\hat{e}(h) = \frac{1}{n+m} \sum_{i=1}^{n+m} \ell(h(x_i), y_i), \tag{1}$$

using the dataset $\{(x_i, y_i)\}_{i=1}^{n+m}$; here $\ell$ measures the mismatch between the prediction $h(x_i)$ and label $y_i$. If $P_t = P_o$, then $e_t(h) - \hat{e}(h) = \mathcal{O}((n+m)^{-1/2})$ (Smola & Schölkopf, 1998). But if $P_t \neq P_o$, then we should expect that error on $P_t$ of a hypothesis obtained by minimizing the average empirical loss can be sub-optimal, especially when the number of OOD samples $m \gg n$.

### 2.1 AN EXAMPLE USING FISHER'S LINEAR DISCRIMINANT

Consider a binary classification problem with one-dimensional inputs in Fig. 1. Target samples are drawn from a Gaussian mixture model (with means $\{-\mu, \mu\}$ for the two classes) and OOD samples are drawn from a Gaussian mixture with means $\{-\mu + \Delta, \mu + \Delta\}$; also see Appendix A.1. Fisher's linear discriminant (FLD) is a linear classifier for such binary classification problems. It computes

$$\hat{h}(x) = \begin{cases} 1, & \omega^\top x > c \\ 0, & \text{otherwise}, \end{cases}$$

where $\omega$ is a projection vector which acts as a feature extractor and $c$ is a threshold that performs one-dimensional discrimination between the two classes. FLD assumes that the class conditional density of each class is a multivariate Gaussian distribution with the same covariance structure. We provide a detailed account of FLD in Appendix A.2.

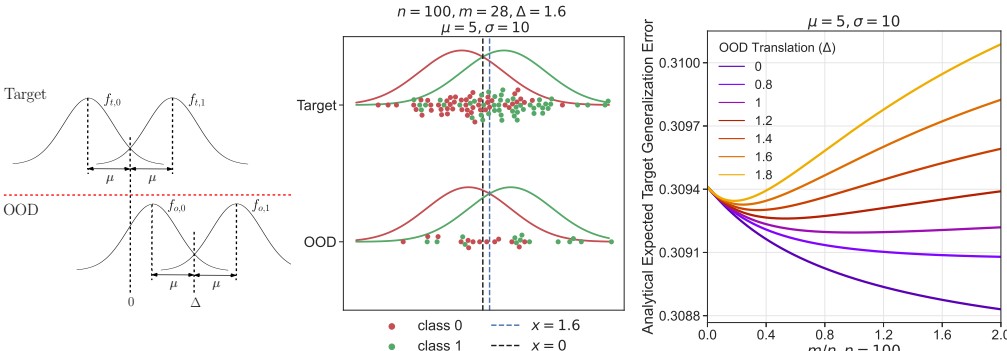

**Figure 1: Left:** A picture of synthetic target and OOD tasks. **Middle:** A schematic of the Gaussian mixture model corresponding to the target task (top) and the OOD samples (bottom). The OOD sample size ($m = 28$) at which the target generalization error is minimized at $\Delta = 1.6$ is indicated at the top. **Right:** For $n = 100$, we plot the generalization error of FLD on the target task as a function of the ratio of OOD and target samples $m/n$, for different types of OOD samples corresponding to different values of $\Delta$. This plot uses the analytical expression for the generalization error in (2); see Appendix A.6 for a numerical simulation study. For small values of $\Delta$, when the two tasks are similar to each other, the generalization error $e_t(h)$ decreases monotonically. However, beyond a certain value of $\Delta$, the generalization error is non-monotonic in the number of OOD samples. The optimal value of $m/n$ which leads to the best generalization error is a function of the relatedness between the two tasks, as governed by $\Delta$ in this example. This non-monotonic behavior can be explained in terms of a bias-variance tradeoff with respect to the target task: a large number of OOD samples reduces the variance but also results in a bias with respect to the optimal hypothesis of the target task.

Suppose we fit a FLD on a dataset which comprises of $n$ target samples and $m$ OOD samples. Also, suppose we do not know which samples are OOD and believe that all the samples in the dataset come from a single target distribution. For univariate data, the FLD decision rule reduces to,

$$\hat{h}(x) = \begin{cases} 1, & x > \frac{\hat{\mu}_0 + \hat{\mu}_1}{2} \\ 0, & \text{otherwise.} \end{cases}$$

Define the decision threshold to be $\hat{c} = (\hat{\mu}_0 + \hat{\mu}_1)/2$. We can calculate (Appendices A.2 and A.3) an analytical expression for the generalization error of FLD on the target task:

$$e_t(\hat{h}) = \frac{1}{2}\left[\Phi\left(\frac{m\Delta - (n+m)\mu}{\sqrt{(n+m)(n+m+1)}}\right) + \Phi\left(\frac{-m\Delta - (n+m)\mu}{\sqrt{(n+m)(n+m+1)}}\right)\right]; \tag{2}$$

here $\Phi$ is the CDF of the standard normal distribution.

Fig. 1 (right) shows how the generalization error $e_t(\hat{h})$ decreases up to some threshold of the ratio between the number of OOD samples and the number of samples from the target task $m/n$ and then increases beyond that. This threshold is different for different values of $\Delta$ as one can see in (2) and Fig. 1 (right). This behavior is surprising because one would *a priori* expect the generalization error to deteriorate monotonically as the number of OOD samples $m$ increases. The fact that such a non-monotonic trend is observed even for a one dimensional Gaussian mixture model and Fisher's Linear Discriminant suggests that this may be a general phenomenon. We can capture this discussion as a theorem with our FLD example above as the proof.

**Theorem 1.** There exist target and OOD tasks, $P_t$ and $P_o$ respectively, such that the generalization error on the target task of the hypothesis that minimizes the empirical risk in (1) is non-monotonic in the number of OOD samples.

**Remark 2 (An intuitive explanation of non-monotonic trends in generalization error).** Suppose that a learning algorithm achieves Bayes optimal error on the target task with high probability when the target sample size $n$ exceeds $N$. We argue that a non-monotonic trend in generalization error is likely to occur when $n < N$, i.e., when target generalization error is higher than the Bayes optimal error. In this case, if we add OOD samples whose empirical distribution is sufficiently close to that of the target task, then this would improve generalization by reducing the variance of the learned hypothesis. But as the OOD sample size increases, the difference in the two distributions becomes apparent and this leads to a bias in the choice of the hypothesis. Fig. 2 illustrates this phenomenon with regards to our FLD example in Fig. 1, by plotting the mean squared error of the decision threshold $\hat{c}$ and its constituent bias and variance components. Roughly speaking, we may understand the non-monotonic trend in generalization as a phenomenon that arises due to the finite number of OOD samples ($m/n$ in the example above). The distance between the distribution of the OOD samples and the distribution of the target task ($\Delta$ in the example) determines the threshold beyond which the error is monotonic. Current tools in learning theory (Smola & Schölkopf, 1998) are fundamentally about understanding generalization when the number of samples is asymptotically large—whether they be from the target task or OOD. In future work, we hope to formally characterize this non-monotonic trend in generalization error by building new learning-theoretic tools.

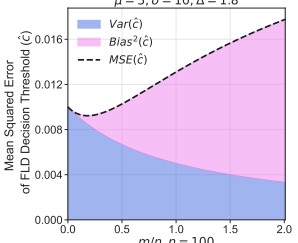

**Figure 2: Mean squared error (MSE) (Y-axis) of the decision threshold $\hat{c}$ of FLD** (see Appendix A.3), for the same setup as that of Fig. 1, plotted against the ratio of the OOD and target samples $m/n$ (X-axis) for $\Delta = 1.8$. Squared bias and variance of the MSE are in violet and blue, respectively. This illustration clearly demonstrates the intuition behind non-monotonic target error: the MSE drops initially because of the smaller variance due to the OOD samples. As more OOD samples are added, MSE increases due to the increasing bias. Non-monotonic trend in MSE of $\hat{c}$ translates to a similar trend in the target generalization error (0-1 loss).

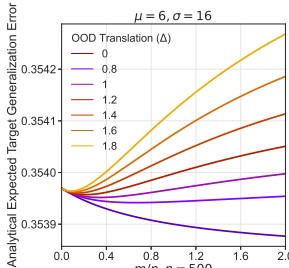

**Figure 3:** We can control the Bayes optimal error by adjusting $\mu, \sigma$ of the Gaussian mixture model in §2.1. As discussed in Remark 2, when the Bayes optimal error is large for ($\mu = 6, \sigma = 16$), we can observe non-monotonic trends even for a large number of target samples ($n = 500$). This suggests that non-monotonic trends in generalization are not limited to small sample sizes.

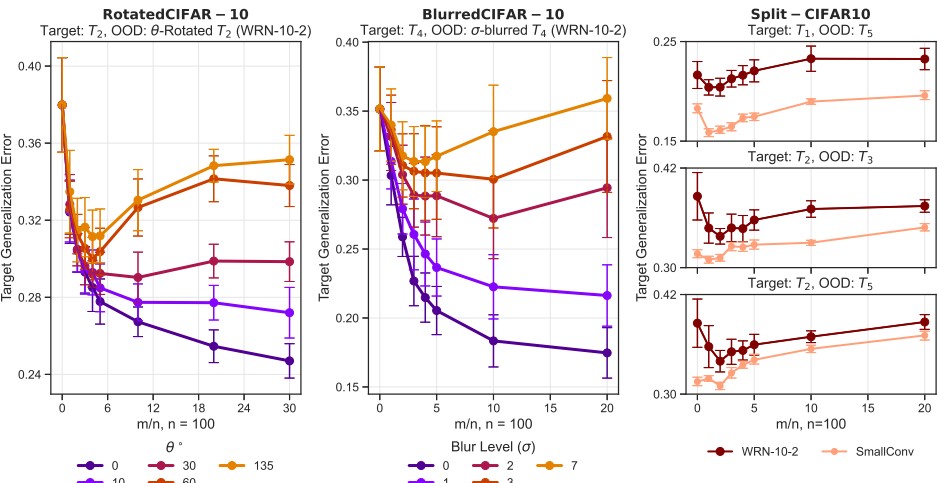

**Figure 4: Left:** Sub-task $T_2$ (Bird vs. Cat) from Split-CIFAR10 is the target task and images of these classes rotated by different angles $\theta^\circ$ are the OOD task. WRN-10-2 architecture was used to train the model. We see non-monotonic curves for larger values of $\theta^\circ$. For $60^\circ$ and $135^\circ$ in particular, the generalization error at $m/n = 20$ is worse than the generalization error with a fewer OOD samples, i.e. OOD samples actively hurt generalization. See Fig. A6 (left) for a similar experiment involving SmallConv, a much smaller CNN. **Middle:** The Split-CIFAR10 binary sub-task $T_4$ (Frog vs. Horse) is the target task and images of these classes subjected to varying levels of Gaussian blur are the OOD samples. WRN-10-2 architecture was used to train the model. Non-monotonic curves are observed for larger levels of blur, while for smaller levels of blur, we notice that adding more OOD data improves the generalization on the target task. **Right:** Generalization error of two separate networks, WRN-10-2 and SmallConv, on the target task is plotted against the number of samples from the OOD task for 3 different pairs of target-OOD tasks from Split-CIFAR10. All the 3 pairs exhibit non-monotonic target generalization trends across both network models. See Appendices B.2 and B.3 for experimental details and Appendix B.7 for experiments on more task pairs (Figs. A4 and A5) and multiple target sample sizes (Fig. A3). Error bars indicate 95% confidence intervals (10 runs).

Even if the non-monotonic trend occurs for relatively small values of target and OOD samples $n$ and $m$ respectively in Fig. 1, this need not always be the case. If the number of samples $N$ required to reach Bayes optimal error in the above remark is large, then a non-monotonic trend could occur even if we have a relatively large target sample size $n$. If $N$ is small, then the non-monotonic trend would not occur typically because then the typical target sample size would be $n > N$. (See Fig. 3)

## 2.2 NON-MONOTONIC TRENDS FOR NEURAL NETWORKS AND REAL DATASETS

We experiment with several popular datasets including MNIST, CIFAR-10, PACS, and DomainNet and 3 different network architectures: (a) a small convolutional network with 0.12M parameters (denoted by *SmallConv*), (b) a wide residual network (Zagoruyko & Komodakis, 2016) of depth 10 and widening factor 2 (WRN-10-2), and (c) a larger wide residual network of depth 16 and widening factor 4 (WRN-16-4). See Appendix B.4 for more details.

**Non-monotonic trend in generalization error can occur due to geometric and semantic nuisances.** Such nuisances are very common even in curated datasets (Van Horn, 2019). We constructed 5 binary classification sub-tasks (denoted by $T_i$ for $i = 1, \ldots, 5$) from CIFAR-10 to study this aspect (see Appendix B.1). We consider a CIFAR-10 sub-task $T_2$ (Bird vs. Cat) as the target and introduce rotated images by a fixed angle between $0^\circ$-$135^\circ$) as OOD samples. Fig. 4 (left) shows that the generalization error decreases monotonically for small rotations but it is non-monotonic for larger angles. Next, we considered the sub-task $T_4$ (Frog vs. Horse) as the target task and generate OOD samples by adding Gaussian blur of varying levels to images from the same task. In Fig. 4 (middle), the generalization error on the target is a monotonically decreasing function of the number of OOD samples for low blur but it increases non-monotonically for high blur.

**Non-monotonic trends can occur when OOD samples are drawn from a different task** Large datasets can contain categories whose appearance evolves in time (e.g., a typical laptop in 2022 looks very different from that of 1992), or categories can have semantic intra-class nuisances (e.g.,

chairs of different shapes). We use CIFAR-10 sub-tasks to study how such differences can lead to non-monotonic trends (see Appendix B.1). For 5 CIFAR-10 sub-tasks; each sub-task is a binary classification problem with two consecutive classes: Airplane vs. Automobile, Bird vs. Cat, etc. We consider $(T_i, T_j)$ as the (target, OOD) task pair and evaluated the trend in generalization error for all 20 distinct pairs of tasks. Fig. 4 (right) illustrates non-monotonic trends for 3 such pairs; see Appendix B for more details.

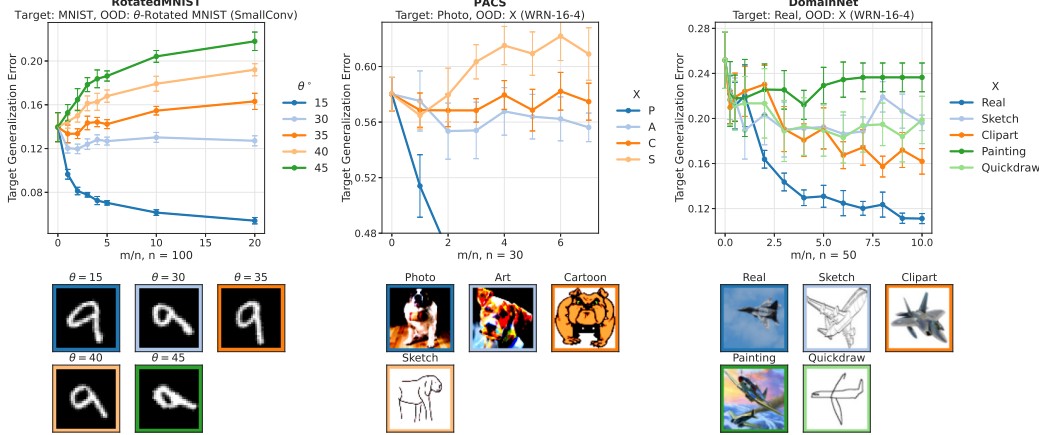

**Figure 5: Non-monotonic trends in target generalization error on three DomainBed benchmarks.** Left: Rotated MNIST (10 classes, 10 target samples/class, *SmallConv*), Middle: PACS (3 classes {dog, elephant, horse}, 10 target samples/class, WRN-16-4), and Right: DomainNet (2 classes {bird, plane}, 25 target samples/class, WRN-16-4). Error bars indicate 95% confidence intervals (10 runs).

**Non-monotonic trends also occur for benchmark domain generalization datasets** We further investigated three widely used benchmarks in the domain generalization literature. First, we consider the Rotated MNIST benchmark from DomainBed (Gulrajani & Lopez-Paz, 2020). We define the 10-way classification of un-rotated MNIST images as the target task and $\theta$-rotated MNIST images as the OOD samples. Similar to the previous rotated CIFAR-10 experiment, we observe non-monotonic trends in target generalization for larger angles $\theta$. Next, we consider the PACS benchmark from DomainBed which contains 4 distinct environments: photo, art, cartoon, and sketch. A 3-way classification task involving photos (real images) is defined as the target task, and we let the corresponding data from other environments be the OOD samples. Interestingly, we observe that when OOD samples consist of sketched images, then the generalization error on the real images exhibits a non-monotonic trend. We also observe similar trends in DomainNet, a benchmark that resembles PACS; see Fig. 5.

**Generalization error is not always non-monotonic even when there is distribution shift** We considered CINIC-10 (Darlow et al., 2018), a dataset which was created by combining CIFAR-10 with

**Figure 6:** Target task is CIFAR-10 and OOD samples are from ImageNet. Although there is a distribution shift that causes the red curve to be higher error than the purple one, there is no non-monotonic trend in the generalization on CIFAR-10 due to OOD samples from ImageNet. Error bars indicate 95% confidence intervals (10 runs).

images selected and down-sampled from ImageNet. We train a network on a subset of CINIC-10 that comprises of both CIFAR-10 and ImageNet images. The target task is CIFAR-10 itself, so images from ImageNet in CINIC-10 act as OOD samples. Fig. 6 demonstrates that having more ImageNet samples in the training data improves the generalization (monotonic decrease) on the target task, but at a slower rate than the instance where the training data is purely comprised of target data. This phenomenon is also demonstrated in Fig. 1: for sufficiently small shifts, the target generalization error decreases as the number of OOD samples increases.

**Effect of pre-training, data-augmentation and hyperparameter optimization** When we do not know which samples in the training data are OOD, we do not have a lot of options to mitigate the deterioration due to the OOD samples. We could employ data augmentations, hyperparameter

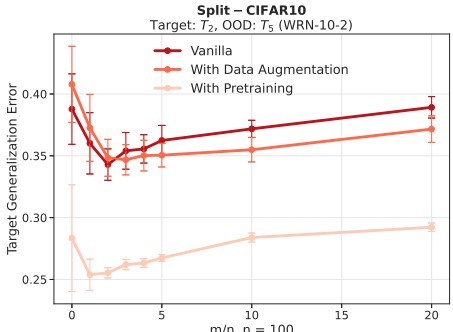 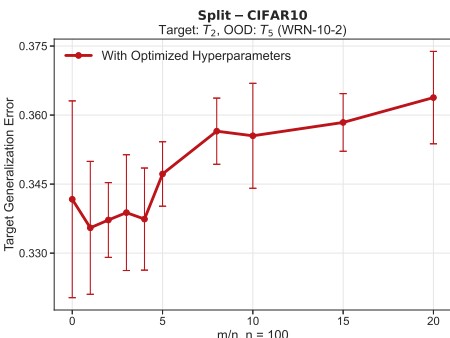

**Figure 7: Left:** Considering CIFAR-10 subtask $T_2$ (Bird vs Cat) as target and $T_5$ (Ship vs Truck) as OOD, we train a WRN-10-2 network with class-balanced datasets containing fixed number of target samples ($n = 100$) and varying number ($m$) of OOD samples, under the following settings: (1) Vanilla — without any data-augmentation or pre-training (darkest shade of red), (2) Data augmentations by introducing padding with random cropping and random left/right flipping (intermediate shade of red), and (3) Pre-training followed by fine-tuning (lightest shade of red). We pre-train the network on 14000 class-balanced ImageNet images from CINIC-10 (see Appendix B.1) belonging to Bird and Cat classes which correspond to our hypothetical target task. Pre-training is performed for 100 epochs with a learning rate of 0.01. Next, we employ a two-step strategy of linear probing (first 50 epochs) and full-fine tuning (last 50 epochs) inspired by Kumar et al. (2022) at a reduced learning rate of 0.001. Note that this fine-tuning is performed on the combined dataset of $n$ target and $m$ OOD samples. Even though data augmentation and pre-training followed by fine-tuning reduce the overall error, we still observe the deterioration of target generalization error as the OOD sample fraction of the dataset increases. **Right:** For each run at each value of $m$, we perform hyperparameter tuning using Ray (Liaw et al., 2018) over a validation set including *only* target samples, and record the target generalization error of the model using the best set of hyperparameters. However, we still observe the target generalization error degrading as the OOD samples increase. **Note that hyperparameter tuning cannot be implemented in reality because we may not know the identity of the target and OOD samples**. So the fact that the non-monotonic trend persists in the hypothetical instance where we *know* the sample identities guarantees that it will occur in practice as well. Error bars indicate 95% confidence intervals over 10 experiments.

optimization, or say pre-training followed by fine-tuning. The second option is difficult to implement for a real problem because the validation data that will be used for hyperparameter optimization will itself have to be drawn from the curated dataset.

We tested whether these three techniques above improve generalization on the target task in the presence of OOD samples. We consider CIFAR-10 sub-task $T_2$ (Bird vs. Cat) as the target task and $T_5$ (Ship vs. Truck) as the OOD task and train a WRN-10-2 network under various settings. The effect of these techniques are reported in Fig. 7 and we find that neither of them manages to curb the deterioration of target generalization error as the OOD sample size of the dataset increases.

## 3 Can we exploit the non-monotonic trend in the generalization error?

Suppose we knew which samples in our dataset were OOD for the target task. Then we should be able to not only mitigate the non-monotonic nature of the generalization error but also exploit it.

**Theorem 3 (Paraphrased from Ben-David et al. (2010)).** For two tasks $P_t$ and $P_o$, let $\hat{h}_\alpha$ be the minimizer of the $\alpha$-weighted empirical risk $\hat{e}_\alpha(h) = \alpha\hat{e}_t(h) + (1-\alpha)\hat{e}_o(h)$ where $\hat{e}_t(h)$ and $\hat{e}_o(h)$ are the empirical risks of $P_t$ and $P_o$ respectively. The generalization error

$$e_t(\hat{h}_\alpha) \le e_t(h_t^*) + 4\sqrt{\left(\frac{\alpha^2}{n} + \frac{(1-\alpha)^2}{m}\right)}\sqrt{V_H - \log\delta} + 2(1-\alpha)d_H(P_t, P_o),$$

with probability at least $1 - \delta$. Here $h_t^* = \operatorname{argmin}_{h \in H} e_t(h)$ is the target error minimizer; $V_H$ is a constant proportional to the VC-dimension of the hypothesis class $H$ and $d_H(P_t, P_o)$ is a notion of relatedness between the tasks $P_t$ and $P_o$.

In other words, if we use an appropriate value of $\alpha$ that makes the second and third terms on the right-hand side small, then we can mitigate the deterioration of generalization error due to OOD samples. If the OOD samples are very different from those of the target task, i.e., if $d(P_t, P_o)$ is large,

then this theorem suggests that we should pick an $\alpha \approx 1$. Doing so effectively ignores the OOD samples and the generalization error then decreases monotonically as $\mathcal{O}(n^{-1/2})$.

### 3.1 CHOOSING THE OPTIMAL $\alpha^*$

If we define $\rho = \frac{\sqrt{V_H - \log \delta}}{d_H(P_t, P_o)}$ to be, roughly speaking, the ratio of the capacity and the distance between tasks, then a short calculation shows that for $\alpha \in [0, 1]$,

$$\alpha^* = \begin{cases} 1 & \text{if } n \geq 4\rho^2, \\ \frac{n}{n+m}\left(1 + \sqrt{\frac{m^2}{4\rho^2(n+m)-nm}}\right) & \text{else.} \end{cases}$$

This suggests that if we have a hypothesis space with small VC-dimension or if the OOD samples and target samples come from very different distributions, then we should train only on the target samples to obtain optimal error. Otherwise, including the OOD samples after appropriately weighing them using $\alpha^*$ can give a better generalization error.

It is not easy to estimate $\rho$ because it depends upon the VC-dimension of the hypothesis class (Ben-David et al., 2010; Vedantam et al., 2021). But in general, we can treat $\alpha$ as a hyperparameter and use validation data to search for its optimal value. For our FLD example we can do slightly better: we can calculate the analytical expression for the generalization error for the hypothesis that minimizes the $\alpha$-weighted empirical risk (see Appendices A.4 and A.5) and calculate $\alpha^*$ by numerically evaluating the expression for $\alpha \in [0, 1]$.

Fig. 8 shows that regardless of the number of OOD samples ($m$) and the relatedness between OOD and target tasks ($\Delta$), we can obtain a generalization error that is always better than that of a hypothesis trained without OOD samples. In other words, if we choose $\alpha^*$ appropriately (Fig. 1 corresponds to choosing $\alpha = 1/2$), then we do not suffer from non-monotonic generalization error on the target task.

### 3.2 TRAINING NEURAL NETWORKS USING THE $\alpha$-WEIGHTED OBJECTIVE

In §2.2, for a variety of computer vision datasets, we found that for some pairs of tasks, the generalization error is non-monotonic in the number of OOD samples. We now show that if we knew which samples were OOD, then we can rectify this trend using an appropriate value of $\alpha^*$ to weigh the samples differently. In Fig. 9, we track the test error of the target task for three cases: training is agnostic to the presence of OOD samples (red), the learner knows which samples are OOD and uses an $\alpha = 1/2$ in the weighted risk to train (yellow, we call this "naive"), and when it uses an optimal value of $\alpha$ using grid-search

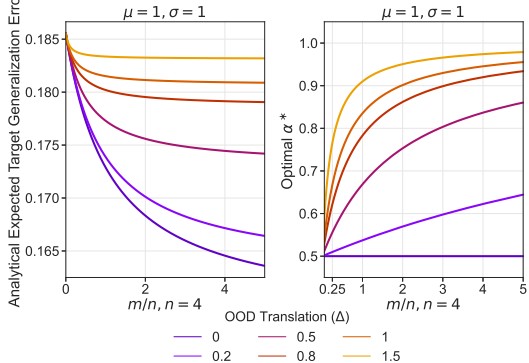

**Figure 8: Left:** Generalization error on the target task for the Gaussian mixture model using a weighted objective (Theorem 3) in FLD; see Appendix A.4. Note that unlike in Fig. 1, the generalization error monotonically decreases with the number of OOD samples $m$. **Right:** The optimal $\alpha^*$ that yields the smallest target generalization error as a function of the number of OOD samples. Note that $\alpha^*$ increases as the number of OOD samples $m$ increases; this increase is more drastic for large values of $\Delta$ and is more gradual for small values of $\Delta$. Observe that $\alpha^* = 1/2$ for all values of $m$ if $\Delta = 0$. See Appendix A.6 for a numerical simulation.

(green). Searching over $\alpha$ improves the test error on all these 3 pairs of target-OOD tasks. See Figs. A4 to A6 for more experiments with similar conclusions.

We also conducted another experiment to check if augmentation can help rectify the non-monotonic trend in the generalization error, using the $\alpha$-weighted objective, i.e., when we know which samples are OOD. As shown in Fig. 10, in this case even naively weighing the objective ($\alpha = 1/2$, yellow) can rectify the non-monotonic trend, using the optimal $\alpha^*$ (green) further improves the error. *This suggests that augmentation is an effective way to mitigate non-monotonic behavior, but only if we use the $\alpha$-weighted objective, which requires knowing which samples are OOD.* As we discussed in Fig. 7 if we do not know which samples are OOD, then augmentation does not help.

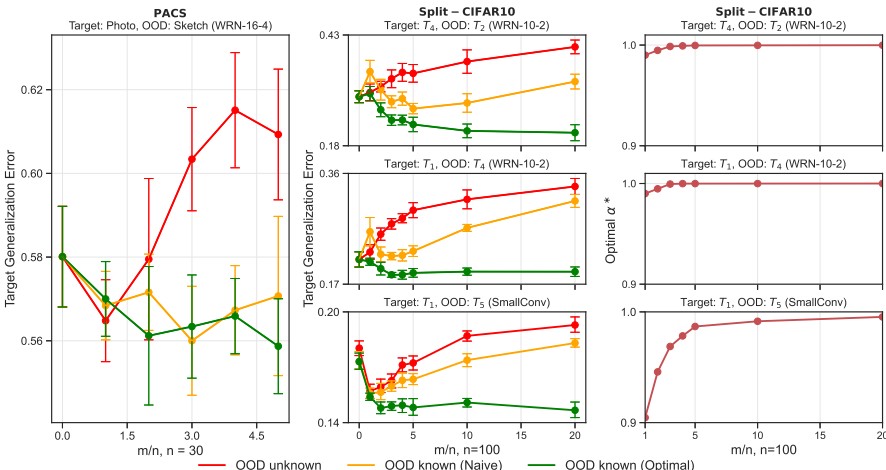

**Figure 9:** Here we present three settings: minimizing the average risk over target and OOD samples is agnostic to OOD samples present (red), minimizing the sum of the average loss of the target and OOD tasks which corresponds to $\alpha = 1/2$ (yellow), minimizing an optimally weighted convex combination of the target and OOD empirical loss (green). The last two settings are only possible when one knows which samples are OOD. For each setting, we plot the generalization error on the target task against the number of OOD samples for (target, OOD) pairs from PACS **(Left)** and CIFAR-10 subtasks **(Middle)**. Unlike in CIFAR-10 task pairs, we observe that in PACS, the target generalization error has a downward trend when $\alpha = 0.5$ (yellow line, left panel). We speculate that this could be due to the similarity between the target and OOD tasks, which causes the model to generalize to the target even at a naive weight. **Right:** The optimal $\alpha^*$ obtained via grid search for the three problems in the middle column plotted against different number of OOD samples. The value of $\alpha^*$ lies very close to 1 but it is never exactly 1. In other words, if we use the weighted objective in Theorem 3 then we always obtain some benefit, even if it is marginal when OOD samples are very different from those of the target. Error bars indicate 95% confidence intervals over 10 experiments.

**Sampling mini-batches during training** For $m \gg n$, mini-batches that are sampled uniformly randomly from the dataset will be dominated by OOD samples. As a result, the gradient even if it is still unbiased, is computed using very few samples from the target task. This leads to an increase in the test error, which is particularly noticeable with $\alpha^*$ chosen appropriately after grid search. We therefore use a biased sampling procedure where each mini-batch contains a fraction $\beta$ samples from the target task and the remainder $1 - \beta$ consists of OOD samples. This parameter controls the bias and variance of the gradient of the target task ($\beta = \frac{n}{n+m}$ gives unbiased gradients with respect to the unweighted total objective and high variance with respect to the target task when $m \gg n$, see Appendix B.5). We found that both $\beta = \{0.5, 0.75\}$ improve test error.

**Weighted objective for over-parameterized networks** It has been argued previously that weighted objectives are not effective for over-parameterized models such as deep networks because both surrogate losses $\hat{e}_t(h)$ and $\hat{e}_o(h)$ are zero when the model fits the training dataset (Byrd & Lipton, 2019). It may therefore seem that the weighted objective in Theorem 3 cannot help us mitigate the non-monotonic nature of the generalization error; indeed the minimizer of $\alpha\hat{e}_t(h) + (1-\alpha)\hat{e}_o(h)$ is the same for any $\alpha$ if the minimum is exactly zero. Our experiments suggest otherwise: the value of $\alpha$ does impact the generalization error—even for deep networks. This is perhaps because even if the cross-entropy loss is near-zero for a deep network towards the end of training, it is never exactly zero.

## 4 RELATED WORK

**Distribution shift** (Quinonero-Candela et al., 2008) and its variants such as covariate shift (Ben-David & Urner, 2012; Reddi et al., 2015), concept drift (Mohri & Muñoz Medina, 2012; Bartlett, 1992; Cavallanti et al., 2007), domain shift (Gulrajani & Lopez-Paz, 2020; Sagawa et al., 2021; Ben-David et al., 2010), sub-population shift (Santurkar et al., 2020; Hu et al., 2018; Sagawa et al., 2019), data poisoning (Yang et al., 2017; Steinhardt et al., 2017), geometric and semantic nuisances (Van Horn, 2019), and flawed annotations (Frénay & Verleysen, 2013) can lead to the presence of OOD samples in a curated dataset, and thereby may yield sub-optimal generalization error on the desired task. While

these problems have primary been studied in the sense of an out-of-domain *distribution*, we believe that we have identified a fundamentally different phenomenon, namely a non-monotonic trend in the generalization error due to the different numbers of OOD samples.

**Internal Dataset Shift** A Recent body of works (Kaplun et al., 2022; Swayamdipta et al., 2020; Siddiqui et al., 2022; Jain et al., 2022) have investigated the presence of noisy, hard-to-learn, and/or negatively influential samples in popular vision benchmarks. Existence of such OOD samples indicates that the internal dataset shift may be a widespread problem in large datasets. Such circumstances may give rise to undesired non-monotonic trends in generalization error, as we have described in our work. This coupled with the difficulty of detecting internal shifts motivates further research into tackling this problem.

**Domain Adaptation** While most works listed above provide attractive ways of adapting or being robust to various modes of shift, a part of our work addresses the question: *if* we know which samples are OOD, then can we optimally utilize them to achieve a better generalization on the desired target task? This is related to domain adaptation (Ben-David et al., 2010; Mansour et al., 2008; Pan et al., 2010; Ganin et al., 2016; Cortes et al., 2019). A large body of work uses weighted-ERM based methods for domain adaptation (Ben-David et al., 2010; Zhang et al., 2012; Blitzer et al., 2007; Bu et al., 2022; Hanneke & Kpotufe, 2019; Redko et al., 2017; Wang et al., 2019; Ben-David et al., 2006); this is either done to address domain shift or to address different distributions of tasks in a transfer or multi-task learning. This body of work is our primary motivation, except that in our case, the "source" task is actually the OOD samples.

**Connection with the theory of domain adaptation** While generalization bounds for weighted-ERM like those of Ben-David et al. (2010) are understood to be meaningful (if not tight; see Vedantam et al. (2021)) for large sample sizes, our work identifies an unusual non-monotonic trend in the generalization error of the target task. First note that the calculations in Ben-David et al. (2010) can be used when we do not know the identity of the OOD samples by setting $\alpha = \frac{n}{n+m}$. One can get an insight into the non-monotonic trend of the target error from Theorem 3 in Ben-David et al. (2010) because the second term

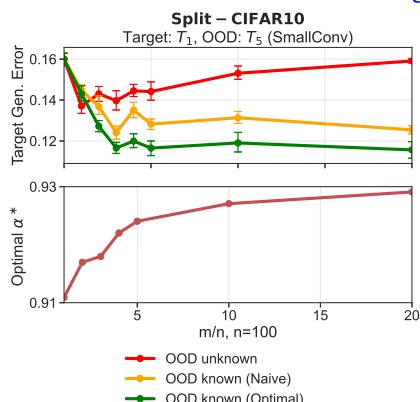

**Figure 10: Effect of data augmentation** (padding with random cropping and random left/right flipping). Although the network trained in the setting where the OOD sample identities are unknown (red) continues to perform poorly with lots of OOD samples, even a naive weighing of the target and OOD loss ($\alpha = 1/2$) is enough to provide a monotonically decreasing error (yellow) when the OOD sample identities are known. This suggests that data augmentation can mitigate some of the anomalies that arise from OOD data, although we can do better by addressing them specifically using, for instance, the weighted objective (green). Error bars indicate 95% confidence intervals over 10 experiments.

$4\sqrt{\left(\frac{\alpha^2}{n} + \frac{(1-\alpha)^2}{m}\right)}\sqrt{V_H - \log \delta}$ decreases with increasing $m$ while the third term $2(1-\alpha)d_H(P_t, P_o)$ increases because $\alpha \to 0$. While such a trend in the upper bound does not directly imply a non-monotonic trend in the error itself, this discussion suggests that the results from our experiments are not inconsistent with existing theory. There is a discrepancy here, e.g., we notice that Ben-David et al. (2010)'s upper bound for naively weighted empirical error ($\alpha = 1/2$) does not have a non-monotonic trend (again, this is only an upper bound on the target error). A more recent paper by Bu et al. (2022) presents an exact characterization of the target generalization error using conditional symmetrized Kullback-Leibler information between the output hypothesis and target samples given the source samples. While they do not identify non-monotonic trends in target generalization error, their tools can potentially be useful to characterize the phenomenon discovered in our work.

**Domain Generalization** seeks to learn a predictor from multiple domains that could perform well on some *unseen* test domain. This unseen test domain can be thought as OOD data. Since no training data is available during the training, the learner needs to make some additional assumptions; one popular assumption is to learn invariances across training and testing domains (Gulrajani & Lopez-Paz, 2020; Arjovsky et al., 2019; Sun & Saenko, 2016). We use several benchmark datasets from this literature, but the goals of this body of work and ours are very different because we are interested only in generalizing on the target task, not generalizing to the domain of the OOD samples.

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

# A  FISHER'S LINEAR DISCRIMINANT (FLD)

## A.1  SYNTHETIC TASKS

The target task $P_t$ and the OOD task $P_o$ are both binary classification problems with one-dimensional inputs. In both tasks, each class is sampled from a univariate Gaussian distribution. The OOD task is the target task translated by $\Delta$. In summary, the target task has the class conditional densities,

$$f_{t,0} \overset{d}{=} \mathcal{N}(-\mu, \sigma^2)$$

$$f_{t,1} \overset{d}{=} \mathcal{N}(+\mu, \sigma^2),$$

while the OOD task distribution has the class conditional densities,

$$f_{o,0} \overset{d}{=} \mathcal{N}(\Delta - \mu, \sigma^2)$$

$$f_{o,1} \overset{d}{=} \mathcal{N}(\Delta + \mu, \sigma^2).$$

We also assume that both the target and OOD tasks have the same label distribution with equal class prior probabilities, i.e. $p(y_t = 1) = p(y_o = 1) = \pi = \frac{1}{2}$. Fig. 1 (left) depicts $P_t$ and $P_o$ pictorially.

## A.2  OOD-AGNOSTIC FISHER'S LINEAR DISCRIMINANT

In this section, we derive FLD when we have samples from a single task – which is also applicable to the OOD-agnostic (when the identity of the OOD samples are not known) setting. Consider a binary classification problem with $D_t = \{(x_i, y_i)\}_{i=1}^n \sim P_t$ where $x_i \in X \subseteq \mathbb{R}^d$ and $y_i \in Y = \{0, 1\}$.

Let $f_k$ and $\pi_k$ be the conditional density and prior probability of class $k$ ($k \in \{0, 1\}$) respectively. The probability that $x$ belongs to class $k$ is

$$p(y = k \mid x) = \frac{\pi_k f_k(x)}{\pi_0 f_0(x) + \pi_1 f_1(x)},$$

and the maximum *a posteriori* estimate of the class label is

$$h(x) = \underset{k \in \{0,1\}}{\operatorname{argmax}} \; p(y = k \mid x) = \underset{k \in \{0,1\}}{\operatorname{argmax}} \; \log(\pi_k f_k(x)). \tag{3}$$

Fisher's linear discriminant (FLD) assumes that each $f_k$ is a multivariate Gaussian distribution with the same covariance matrix $\Sigma$, i.e,

$$f_k(x) = \frac{1}{(2\pi)^{d/2} |\Sigma|^{1/2}} \exp\left( -\frac{1}{2}(x - \mu_k)^\top \Sigma^{-1}(x - \mu_k) \right).$$

Under this assumption, the joint-density $f$ of $(x, y)$ becomes,

$$f(x, y) \propto \prod_{k=0}^{1} \left[ \frac{\pi_k}{|\Sigma|^{1/2}} \exp\left( -\frac{1}{2}(x - \mu_k)^\top \Sigma^{-1}(x - \mu_k) \right) \right]^{\mathbf{1}[y=k]}$$

Therefore, the log-likelihood $l(\mu_0, \mu_1, \Sigma, \pi_0, \pi_1)$ over $D_t$ is given by,

$$\ell(\mu_0, \mu_1, \Sigma, \pi_0, \pi_1) = \sum_{k=0}^{1} \sum_{(x,y) \in D_{t,k}} \left[ \log \pi_k - \frac{1}{2} \log |\Sigma| - \frac{1}{2}(x - \mu_k)^\top \Sigma^{-1}(x - \mu_k) \right] + \text{const.}$$

where $D_{t,k}$ is the set of samples of $D_t$ that belongs to class $k$. Based on the likelihood function above, we can obtain the maximum likelihood estimates $\hat{\mu}_k, \hat{\Sigma}, \hat{\pi}_k$. The expression for the estimate $\hat{\mu}_k$ is

$$\hat{\mu}_k = \frac{1}{|D_{t,k}|} \sum_{(x,y) \in D_{t,k}} x. \tag{4}$$

Plugging these estimates into (3), we get,

$$\hat{h}(x) = \underset{k \in \{0,1\}}{\operatorname{argmax}} \left[ \log \hat{\pi}_k - \frac{1}{2} \log |\hat{\Sigma}| - \frac{1}{2}(x - \hat{\mu}_k)^\top \hat{\Sigma}^{-1}(x - \hat{\mu}_k) \right]$$

$$= \underset{k \in \{0,1\}}{\operatorname{argmax}} \left[ \log \hat{\pi}_k - \frac{1}{2} \log |\hat{\Sigma}| + x^\top \hat{\Sigma}^{-1} \hat{\mu}_k - \frac{1}{2} \hat{\mu}_k^\top \hat{\Sigma}^{-1} \mu_k \right]$$

Therefore, $\hat{h}(x) = 1$ iff,

$$x^\top \hat{\Sigma}^{-1} \hat{\mu}_1 - \frac{1}{2} \hat{\mu}_1^\top \hat{\Sigma}^{-1} \mu_1 + \log \hat{\pi}_1 > x^\top \hat{\Sigma}^{-1} \hat{\mu}_0 - \frac{1}{2} \hat{\mu}_0^\top \hat{\Sigma}^{-1} \mu_0 + \log \hat{\pi}_0$$

$$x^\top \hat{\Sigma}^{-1} \hat{\mu}_1 - x^\top \hat{\Sigma}^{-1} \hat{\mu}_0 > \frac{1}{2} \hat{\mu}_1^\top \hat{\Sigma}^{-1} \mu_1 - \frac{1}{2} \hat{\mu}_0^\top \hat{\Sigma}^{-1} \mu_0 + \log \hat{\pi}_0 - \log \hat{\pi}_1$$

$$(\hat{\Sigma}^{-1}(\hat{\mu}_1 - \hat{\mu}_0))^\top x > (\hat{\Sigma}^{-1}(\hat{\mu}_1 - \hat{\mu}_0))^\top \left( \frac{\hat{\mu}_0 + \hat{\mu}_1}{2} \right) + \log \frac{\hat{\pi}_0}{\hat{\pi}_1}$$

Hence the FLD decision rule $\hat{h}(x)$ is

$$\hat{h}(x) = \begin{cases} 1, & \omega^\top x > c \\ 0, & \text{otherwise} \end{cases}$$

where $\omega = \hat{\Sigma}^{-1}(\hat{\mu}_1 - \hat{\mu}_0)$ is a projection vector and $c = \omega^\top \left( \frac{\hat{\mu}_0 + \hat{\mu}_1}{2} \right) + \log \frac{\hat{\pi}_0}{\hat{\pi}_1}$ is a threshold. When $d = 1$ and $\pi_0 = \pi_1$, the decision rule reduces to

$$\hat{h}(x) = \begin{cases} 1, & x > \frac{\hat{\mu}_0 + \hat{\mu}_1}{2} \\ 0, & \text{otherwise} \end{cases} \tag{5}$$

## A.3 Deriving the Generalization Error of the Target Task for Synthetic Tasks with FLD

We would like to derive an expression for the average generalization error of the target task, when we consider the synthetic tasks described in Appendix A.1. For simplicity, we set the variance $\sigma^2$ of the class conditional densities of the synthetic tasks to $1$.

In the OOD-agnostic setting, the learning algorithm sees a single dataset $D = D_t \cup D_o$ of size $n + m$ which is a combination of both target and OOD samples. We can estimate $\mu_k$ using (4) to obtain

$$\hat{\mu}_k = \frac{1}{|D_k|} \sum_{(x,y) \in D_k} x = \frac{\sum_{(x,y) \in D_{t,k}} x + \sum_{(x,y) \in D_{o,k}} x}{n_k + m_k}$$

$$= \frac{n_k \bar{x}_{t,k} + m_k \bar{x}_{o,k}}{n_k + m_k} \tag{6}$$

$$= \frac{n \bar{x}_{t,k} + m \bar{x}_{o,k}}{n + m}.$$

where $D_k$ is the set of samples of $D$ that belongs to class $k$, $n_k = |D_{t,k}|$ and $m_k = |D_{o,k}|$ for $k \in \{0, 1\}$. $\bar{x}_{t,k}$ and $\bar{x}_{o,k}$ denote the sample means of class $k$ in target and OOD datasets respectively. We assume that $\pi = \frac{1}{2}$ from which it follows that $n_k = n\pi_k = \frac{n}{2}$ and $m_k = m\pi_k = \frac{m}{2}$. We cannot explicitly compute $\bar{x}_{t,k}$ and $\bar{x}_{o,k}$ when the OOD samples are not explicitly known, because we cannot separate target samples from OOD samples in $D$.

Since the samples are drawn from Gaussians, their averages also follow Gaussian distributions. Hence, the threshold $\hat{c} = \frac{\hat{\mu}_0 + \hat{\mu}_1}{2}$ of the hypothesis $\hat{h}$, estimated using FLD, is a random variable with a Gaussian distribution i.e., $\hat{c} \sim \mathcal{N}(\mu_h, \sigma_h^2)$ where

$$\mu_h = \mathbb{E}[\hat{c}] = \frac{m\Delta}{n + m},$$

$$\sigma_h^2 = \text{Var}[\hat{c}] = \frac{1}{n + m}.$$

The target error of a hypothesis $\hat{h}$ is

$$p(\hat{h}(x) \neq y \mid x, \hat{c}) = \frac{1}{2} p_{x \sim f_{t,1}}[x < \hat{c}] + \frac{1}{2} p_{x \sim f_{t,0}}[x > \hat{c}]$$

$$= \frac{1}{2} + \frac{1}{2} p_{x \sim f_{t,1}}[x < \hat{c}] - \frac{1}{2} p_{x \sim f_{t,0}}[x < \hat{c}]$$

$$= \frac{1}{2} \left[ 1 + \Phi(\hat{c} - \mu) - \Phi(\hat{c} + \mu) \right] \tag{7}$$

Using (7), the expected error on the target task $e_t(\hat{h}) = \mathbb{E}_{\hat{c} \sim \mathcal{N}(\mu_h, \sigma_h^2)}[p(\hat{h}(x) \neq y \mid x, \hat{c})]$ is given by,

$$
\begin{aligned}
e_t(\hat{h}) &= \int_{-\infty}^{\infty} \frac{1}{2}\left[1 + \Phi(\hat{c} - \mu) - \Phi(\hat{c} + \mu)\right] \frac{1}{\sigma_h} \phi\left(\frac{\hat{c} - \mu_h}{\sigma_h}\right) d\hat{c} \\
&= \int_{-\infty}^{\infty} \frac{1}{2}\left[1 + \Phi(y\sigma_h + \mu_h - \mu) - \Phi(y\sigma_h + \mu_h + \mu)\right] \phi(y) dy \\
&= \frac{1}{2}\left[\Phi\left(\frac{\mu_h - \mu}{\sqrt{1 + \sigma_h^2}}\right) + \Phi\left(\frac{-\mu_h - \mu}{\sqrt{1 + \sigma_h^2}}\right)\right]
\end{aligned}
$$

In the last equality, we make use of the identity $\int_{-\infty}^{\infty} \Phi(cx + d)\phi(x)dx = \Phi\left(\frac{d}{\sqrt{1+c^2}}\right)$ where $\phi$ and $\Phi$ are the PDF and CDF of the standard normal. Substituting the expressions for $\mu_h, \sigma_h^2$ into the above equation, we get

$$
e_t(\hat{h}) = \frac{1}{2}\left[\Phi\left(\frac{m\Delta - (n+m)\mu}{\sqrt{(n+m)(n+m+1)}}\right) + \Phi\left(\frac{-m\Delta - (n+m)\mu}{\sqrt{(n+m)(n+m+1)}}\right)\right] \tag{8}
$$

For synthetic tasks with $\sigma^2 \neq 1$, the target generalization error can be obtained by simply replacing $\mu$ and $\Delta$ with $\frac{\mu}{\sigma}$ and $\frac{\Delta}{\sigma}$ respectively in (8).

## A.4 OOD-Aware Weighted Fisher's Linear Discriminant

We consider a target dataset $D_t = \{(x_i, y_i)\}_{i=1}^n$ and an OOD dataset $D_o = \{(x_i, y_i)\}_{i=1}^m$, which are samples from the synthetic tasks from Appendix A.1. This setting differs from Appendix A.3 since we know whether each sample from $D = D_t \cup D_o$ is OOD or not. This difference allows us to consider a log-likelihood function that weights the target and OOD samples differently, i.e. we consider

$$
\ell(\mu_0, \mu_1, \sigma_0^2, \sigma_1^2) = \sum_{k=0}^{1}\left(\alpha \sum_{(x,y) \in D_{t,k}}\left[-\log \sigma_k - \frac{(x - \mu_k)^2}{2\sigma_k^2}\right] + (1 - \alpha) \sum_{(x,y) \in D_{o,k}}\left[-\log \sigma_k - \frac{(x - \mu_k)^2}{2\sigma_k^2}\right]\right) + \text{const.}. \tag{9}
$$

$\alpha$ is a weight that controls the contribution of the OOD samples in the log-likelihood function. Under the above log-likelihood, the maximum likelihood estimate for $\mu_k$ is

$$
\hat{\mu}_k = \frac{\alpha \sum_{(x,y) \in D_{t,k}} x + (1 - \alpha) \sum_{(x,y) \in D_{o,k}} x}{\alpha |D_{t,k}| + (1 - \alpha)|D_{o,k}|}. \tag{10}
$$

We can make use of the above $\hat{\mu}_k$ to get a weighted FLD decision rule using (5).

## A.5 Deriving the Generalization Error of the Target Task for Synthetic Tasks with Weighted FLD

We consider the synthetic tasks in Appendix A.1 with $\sigma^2 = 1$. We re-write $\hat{\mu}_k$ from (10) using notation from Appendix A.3:

$$
\hat{\mu}_k = \frac{n\alpha \bar{x}_{t,k} + m(1 - \alpha)\bar{x}_{o,k}}{n\alpha + m(1 - \alpha)}.
$$

We can explicitly compute $\bar{x}_{t,k}$ and $\bar{x}_{o,k}$ in the OOD-aware setting since we can separate target samples from OOD samples. For the synthetic dataset, the threshold $\hat{c}_\alpha = \frac{\hat{\mu}_0 + \hat{\mu}_1}{2}$ of the hypothesis $\hat{h}_\alpha$ follows a normal distribution $\mathcal{N}(\mu_{h\alpha}, \sigma_{h\alpha}^2)$ where

$$
\mu_{h\alpha} = \mathbb{E}[\hat{c}_\alpha] = \frac{m(1 - \alpha)\Delta}{n\alpha + m(1 - \alpha)}
$$

$$
\sigma_{h\alpha}^2 = \text{Var}[\hat{c}_\alpha] = \frac{\alpha^2 n + (1 - \alpha)^2 m}{(\alpha n + (1 - \alpha)m)^2}
$$

Similar to the Appendix A.3, we derive an analytical expression for the expected target risk of the weighted FLD, which is

$$
e_t(\hat{h}_\alpha) = \frac{1}{2}\left[\Phi\left(\frac{\mu_{h\alpha} - \mu}{\sqrt{1 + \sigma_{h\alpha}^2}}\right) + \Phi\left(\frac{-\mu_{h\alpha} - \mu}{\sqrt{1 + \sigma_{h\alpha}^2}}\right)\right] \tag{11}
$$

## A.6 Additional Experiments using FLD

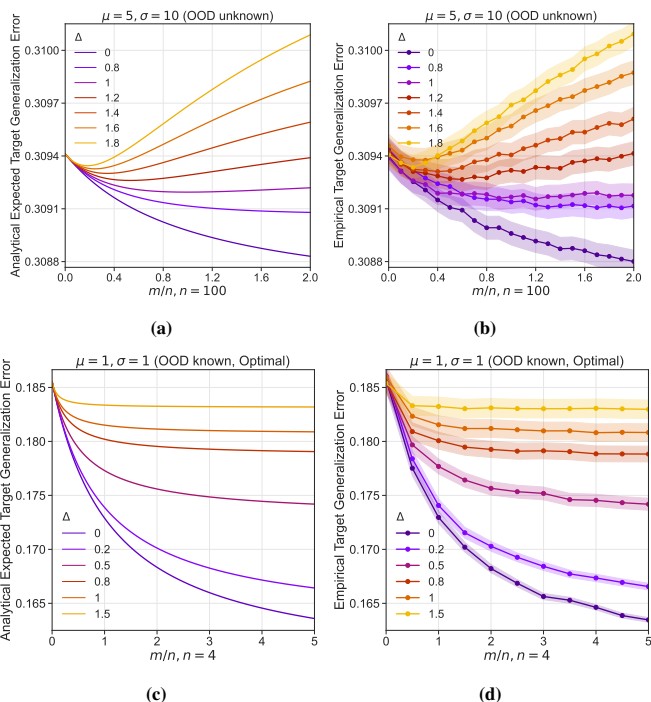

**Figure A1:** The FLD generalization error (Y-axis) on the target task is plotted against the ratio of OOD samples to target samples (X-axis). Figures (a) and (c) are plotted using the analytical expressions in (8) and (11) respectively while figures (b) and (d) are the corresponding plots from Monte-carlo simulations. The Monte-carlo simulations agree with the plots from the analytical expression, which validates its correctness. **(a) and (b):** The figure is identical to Fig. 1 and considers synthetic tasks with $n = 100$, $\mu = 5$ and $\sigma = 10$ in the OOD-agnostic setting. While a small number of OOD samples improves generalization on the target task, lots of samples increase the generalization error on the target task. **(c) and (d):** The figures consider synthetic tasks with $n = 4$, $\mu = 1$ and $\sigma = 1$ in the OOD-aware setting. If we consider the weighted FLD trained with optimal $\alpha^*$, then the average generalization error monotonically decreases with more OOD samples. Shaded regions indicate 95% confidence intervals over the Monte-Carlo replicates.

# B Experiments with Neural Networks

## B.1 Datasets

We experiment on images from CIFAR-10, CINIC-10 (Darlow et al., 2018) and several datasets from the DomainBed benchmark (Gulrajani & Lopez-Paz, 2020): Rotated MNIST (Ghifary et al., 2015), PACS (Li et al., 2017), and DomainNet (Peng et al., 2019). We construct sub-tasks from these datasets as explained below.

**CIFAR-10**  We use of tasks from Split-CIFAR10 (Zenke et al., 2017) which are five binary classification sub-tasks constructed by grouping consecutive labels of CIFAR-10. The 5 tasks are airplane vs. automobile ($T_1$), bird vs. cat ($T_2$), deer vs. dog ($T_3$), frog vs. horse ($T_4$) and ship vs truck ($T_5$). All the images are of size $(3, 32, 32)$.

**CINIC-10**  This dataset combines CIFAR-10 with downsampled images from ImageNet. It contains images of size $(3, 32, 32)$ across 10 classes (same classes as CIFAR-10). As there are two sources of the images within this dataset, it is a natural candidate for studying distribution shift. The construction of the dataset motivates us to consider two tasks from CINIC-10: (1) Task with only CIFAR images, and (2) Task with only ImageNet images.

**Rotated MNIST**  This dataset is constructed from MNIST by rotating the images (which are of size $(1, 28, 28)$. All MNIST images rotated by an angle $\theta^\circ$ are considered to belong to the same

task. Hence, we can consider the family of tasks which is characterized by 10-way classification of hand-written digit images rotated $\theta^\circ$. By varying $\theta$, we can obtain a number of different tasks.

**PACS**   PACS contains images of size $(3, 224, 244)$ with 7 classes present across 4 domains {art, cartoons, photos, sketches}. In our experiments, we consider only 3 classes ({Dog, Elephant, Horse}) out of the 7 and consider the 3-way classification of images from a given domain as a task. Therefore, we can have a total of 4 distinct tasks from PACS.

**DomainNet**   Similar to PACS, this dataset contains images of size $(3, 224, 244)$ from 6 domains { clipart, infograph, painting, quickdraw, real, sketches} across 345 classes. In our experiments, we consider only 2 classes, ({Bird, Plane}) and consider the binary classification of images from a given domain as task. As a result, we can have a total of 6 distinct tasks from PACS.

### B.2   Forming Target and OOD Tasks

We consider two types of setups to study the impact of OOD data:

**OOD data arising due to geometric intra-class nuisances**   We study the effect of intra-class nuisances using a classification task as the target task and transformed versions of the same task as different OOD tasks. In this regard, we consider the following experimental setups.

1. **Rotated MNIST: unrotated-domain task as target and $\theta^\circ$- rotated-domain task as OOD:** We consider the 10-way classification (see Appendix B.1) of unrotated images as the target task and that of the $\theta^\circ$- rotated images as the OOD samples. We can have different OOD tasks by selecting different values for $\theta$.
2. **Rotated CIFAR-10: $T_2$ as target and rotated $T_2$ as OOD:** We choose the bird vs. cat ($T_2$) task from Split-CIFAR10 as the target task. We then rotate the images of $T_2$ by an angle $\theta^\circ$ counter-clockwise around their centers to form a new task denoted by $\theta$-$T_2$, which we consider as the OOD task. Different OOD tasks can be obtained by selecting different values for $\theta$.
3. **Blurred CIFAR-10: $T_4$ as target and blurred $T_4$ as OOD:** We choose the Frog vs. Horse ($T_4$) task from Split-CIFAR10 as the target task. We then add Gaussian blur with standard deviation $\sigma$ to the images of $T_4$ to form a new task denoted by $\sigma$-$T_2$, which we consider as the OOD task. By setting distinct values for $\sigma$, we can have different OOD tasks.

**OOD data arising due to category shifts and concept drifts**   We study this aspect using two different target and OOD classification problems as described below.

1. **Split-CIFAR10: $T_i$ as Target and $T_j$ as OOD:** We choose a pair of distinct tasks from the 5 binary classification tasks of Split-CIFAR10 and consider one as the target task and the other as the OOD task. We perform experiments for all pairs of tasks (20 in total) in Split-CIFAR10.
2. **PACS: Photo-domain task as target and X-domain task as OOD:** Out of the four 3-way classification tasks from PACS described in Appendix B.1, we select the photo-domain task as the target task and consider one of the remaining 3 domain tasks (for instance, the sketch-domain task) as the OOD task.
3. **DomainNet: Real-domain task as target and X-domain task as OOD:** Out of the six binary classification tasks from DomainNet described in Appendix B.1, we consider the real-domain task as the target task and select one of the remaining 5 domain tasks (for instance, the painting-domain task) as the OOD task.
4. **CINIC-10: CIFAR task as target and ImageNet task as OOD:** Here we simply select the 10-way classification of CIFAR images as the target task and that of ImageNet as the OOD task.

### B.3   Experimental Details

In the above experiments, for each random seed, we randomly select a fixed sample of size $n$ from the target task. Next, we select samples from the OOD task of varying sizes $m$ such that each progressive sample is a subset of the next sample. The samples from both target and OOD tasks preserve the ratio of the classes. For rotated MNIST, rotated CIFAR-10, and blurred CIFAR-10, when selecting

multiple sets of OOD samples, the OOD images that correspond to the $n$ selected target images are disregarded. For PACS and DomainNet, the images are downsampled to $(3, 64, 64)$ during training.

For both the OOD-agnostic (OOD unknown) and OOD-aware (OOD known) settings, at each $m$-value, we construct a combined dataset containing the $n$ sized target set and $m$ sized OOD set. We use a CNN (see Appendix B.4) for experiments in the both of these settings. We experiment with $\alpha$ fixed to $0.5$ (naive OOD-aware model) and with the optimal $\alpha^*$. We average the runs over 10 random seeds and evaluate on a test set comprised of only target samples.

In the optimal OOD-aware setting, we use a grid-search to find the optimal $\alpha^*$ for each value of $m$. We use an adaptive equally-spaced $\alpha$ search set of size 10 such that it ranges from $\alpha^*_{prev}$ to $1.0$ (excluding $1.0$) where $\alpha^*_{prev}$ is the optimal value of $\alpha$ corresponding to the previous value of $m$. We use this search space since we expect $\alpha^*$ to be an increasing function of $m$.

### B.4 Neural Architectures and Training

We primarily use 3 different network architectures in our experiments: (a) a small convolutional network with 0.12M parameters (denoted by *SmallConv*), (b) a wide residual network (Zagoruyko & Komodakis, 2016) of depth 10 and widening factor 2 (WRN-10-2), and (c) a larger wide residual network of depth 16 and widening factor 4 (WRN-16-4). SmallConv comprises of 3 convolution layers (kernel size 3 and 80 filters) interleaved with max-pooling, ReLU, batch-norm layers, with a fully-connected classifier layer in our experiments.

Table A1 provides a summary of network architectures used in the experiments described earlier. All the networks are trained using stochastic gradient descent (SGD) with Nesterov's momentum and cosine-annealed learning rate. The hyperparameters used for the training are, learning rate of $0.01$, and a weight-decay of $10^{-5}$. All the images are normalized to have mean 0.5 and standard deviation 0.25. In the OOD-agnostic setting, we use sampling without replacement to construct the mini-batches. In the OOD-aware settings (both naive and optimal), we construct mini-batches with a fixed ratio of target and OOD samples. See Appendix B.5 and Fig. A2 for more details.

| Experiment | Network(s) | # classes | n | Image Size | Mini-Batch Size |
|---|---|---|---|---|---|
| Rotated MNIST | SmallConv | 10 | 100 | (1,28,28) | 128 |
| Rotated CIFAR-10 | SmallConv, WRN-10-2 | 2 | 100 | (3,32,32) | 128 |
| Blurred CIFAR-10 | WRN-10-2 | 2 | 100 | (3,32,32) | 128 |
| Split-CIFAR10 | SmallConv, WRN-10-2 | 2 | 100 | (3,32,32) | 128 |
| PACS | WRN-16-4 | 3 | 30 | (3,64,64) | 16 |
| DomainNet | WRN-16-4 | 2 | 50 | (3,64,64) | 16 |
| CINIC-10 | WRN-10-2 | 10 | 100 | (3,32,32) | 128 |

**Table A1:** Summary of network architectures used in the experiments

### B.5 Construction of Mini-Batches

Consider a mini-batch $\{(x_{b_i}, y_{b_i})\}_{i=1}^{B}$ of size $B$. Let the randomly chosen mini-batch contains $B_t$ target samples and $B_o$ OOD samples ($B = B_t + B_o$). Let $\hat{e}_{B,t}(h)$ and $\hat{e}_{B,o}(h)$ denote the average mini-batch surrogate losses for the $B_t$ target samples and $B_o$ OOD samples respectively.

In the OOD-aware (when we know which samples are OOD) setting, $\hat{e}_{B,t}(h)$ and $\hat{e}_{B,o}(h)$ can be computed explicitly for each mini-batch resulting in the mini-batch gradient

$$\hat{\nabla}\hat{e}_B(h) = \alpha\hat{\nabla}\hat{e}_{B,t}(h) + (1-\alpha)\hat{\nabla}\hat{e}_{B,o}(h). \tag{12}$$

If we were to sample without replacement, we expect the fraction of the target samples in every mini-batch to approximately equal $\frac{n}{n+m}$ on average. However, if $m >> n$, we run into a couple of issues. First, we observe that most mini-batches have no target samples, making it impossible to compute $\hat{\nabla}\hat{e}_{B,t}(h)$. Next, even if the mini-batch does have some target samples, there are very few of them, resulting in high variance in the estimate $\hat{\nabla}\hat{e}_{B,t}(h)$.

Hence, we find it beneficial to consider alternative sampling schemes for the mini-batch. Independent of the values of $n$ and $m$, we use a sampler which ensures that every mini-batch has a fixed fraction of target samples, which we denote by $\beta$. For example if the mini-batch size $B$ is 20 and if

$\beta = 0.5$, then every mini-batch has 10 target samples and 10 OOD samples regardless of $n$ and $m$. Note that this sampling biases the gradient, but results in reduced variance estimates. In practice, we observe improved test errors when we set $\beta$ to either 0.5 or 0.75.

## B.6 COMPARING THE EFFECT OF USING CONVENTIONAL AND CUSTOM BATCHES

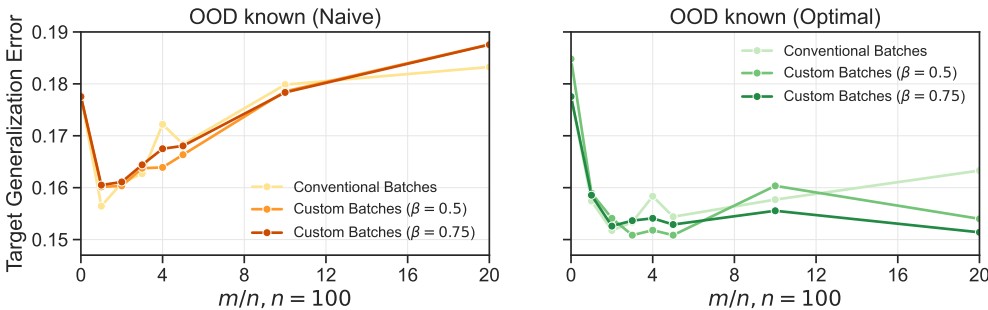

**Figure A2:** The test error of a neural network (SmallConv) on the target task (Y-axis) is plotted against the number of samples from the OOD task (X-axis) for the target-OOD task pair of $T_1$ and $T_5$. One set of curves (lightest shade of green and yellow) considers mini-batches which are constructed using sampling without replacement; This is the conventional strategy used in supervised learning. The other curves consider $\beta = 0.5$ (intermediate shades of orange and green) and $\beta = 0.75$ (darkest shade of red and green). All plots are in the OOD-aware setting. **Left:** If we consider $\alpha = 0.5$, then the choice of $\beta$ has little effect on the generalization error. **Right:** However, if we use the $\alpha^*$ to weight the OOD and target losses, then the generalization error depends on the the choice of $\beta$ with $\beta = 0.75$ having the lowest test error.

## B.7 ADDITIONAL EXPERIMENTS WITH NEURAL NETWORKS

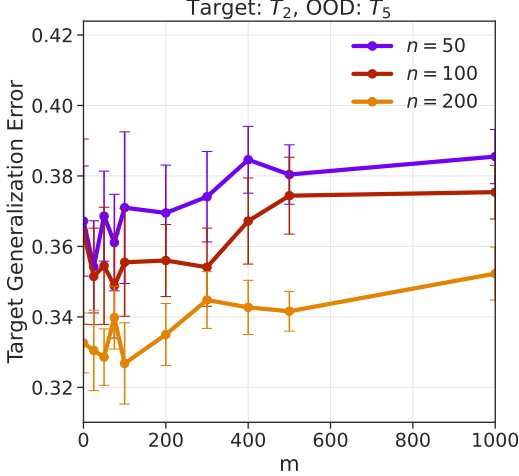

**Figure A3:** We plot the generalization error on the target task (Y-axis) against the number of samples $m$ (X-axis) from the OOD task across three different target sample sizes, $n = 50, 100$ and $200$ for the target-OOD task pair $T_2$ and $T_5$ from Split-CIFAR10. Non-monotonic trends in generalization error are present in all the three cases. The trend is less apparent for $n = 50$ since the number of samples is small resulting in a large variance. Error bars indicate 95% confidence intervals (10 runs).

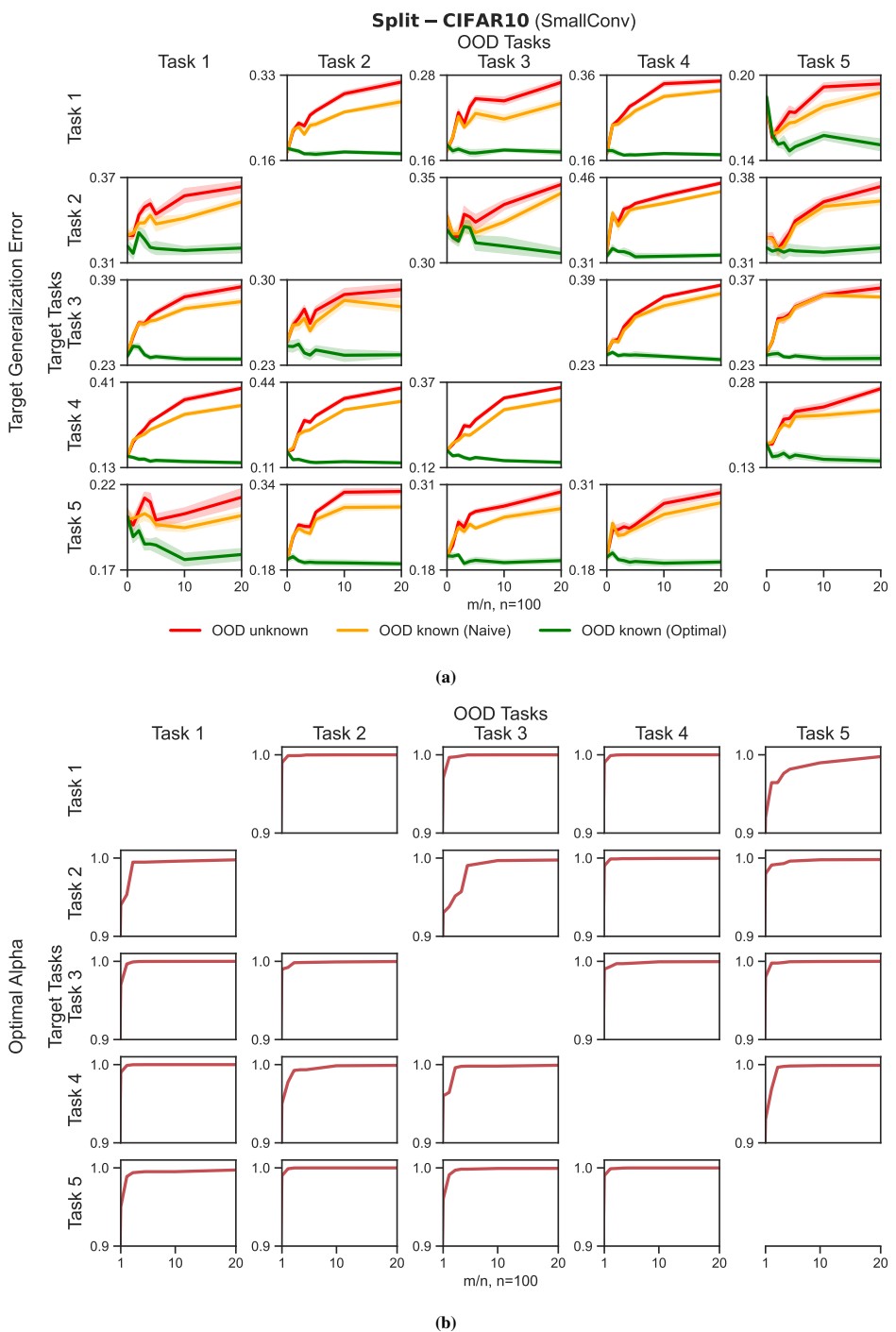

**Figure A4: (a)** We plot the test error of SmallConv on the target task (Y-axis) against the ratio of number of samples from the OOD task to the number of samples on the target task (X-axis), for all target-OOD task pairs from Split-CIFAR10. A neural net trained with a loss weighted by $\alpha^*$ is able to leverage OOD data to improve the networks ability to generalize on the target task. Shaded regions indicate 95% confidence intervals over 10 experiments. **(b)** The optimal $\alpha^*$ (Y-axis) is plotted against the number of OOD samples (X-axis) for the optimally weighted OOD-aware setting. As we increase the number of OOD samples, we see that $\alpha^*$ increases. This allows us to balance the variance from few target samples and the bias from using OOD samples from a different disitribution.

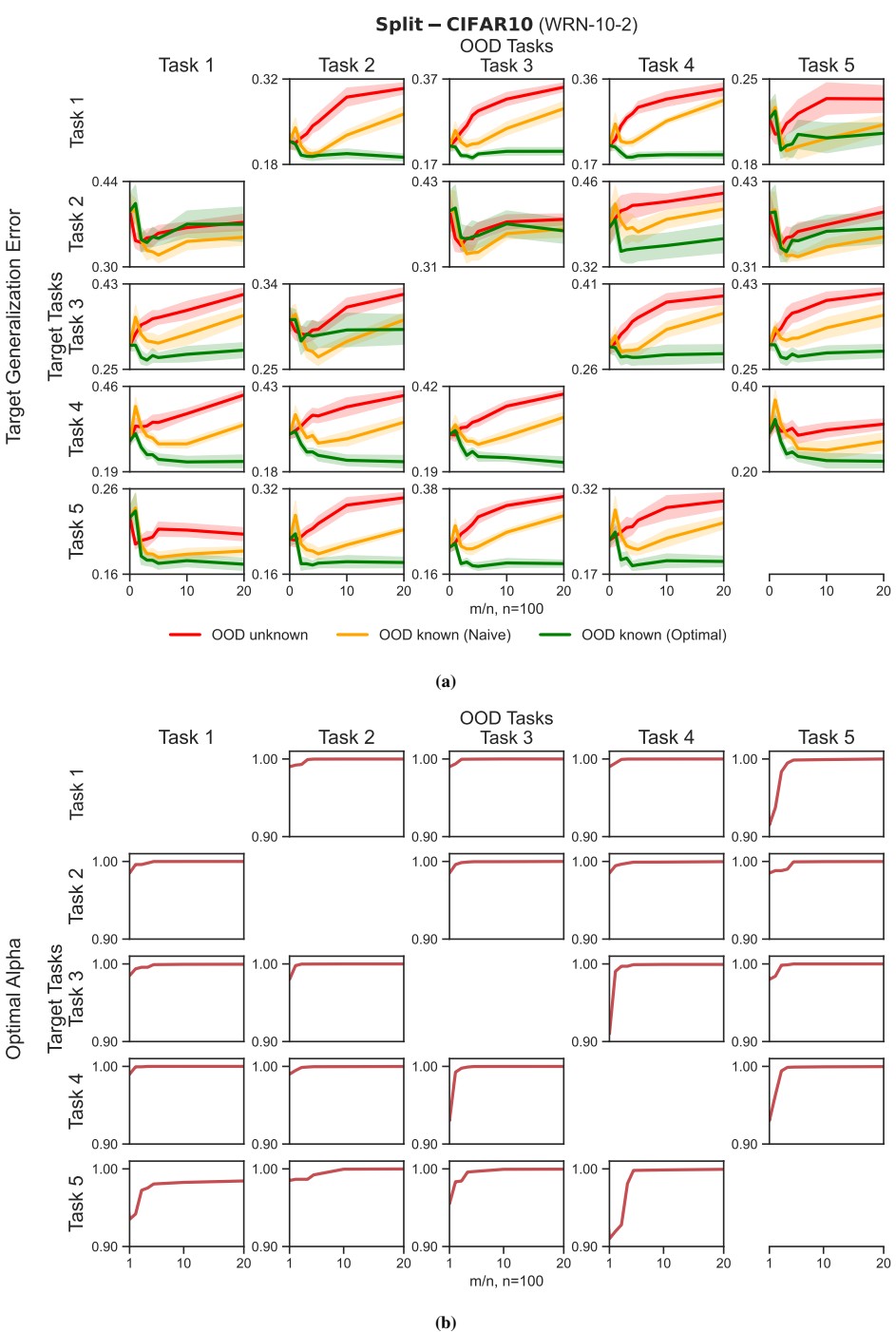

**Figure A5:** **(a)** We plot the test error of WRN-10-2 on the target task (Y-axis) against the ratio of number of samples from the OOD task to the number of samples on the target task (X-axis), for all target-OOD task pairs from Split-CIFAR10. A neural net trained with a loss weighted by $\alpha^*$ is able to leverage OOD data to improve the networks ability to generalize on the target task. Shaded regions indicate 95% confidence intervals over 10 experiments. **(b)** The optimal $\alpha^*$ (Y-axis) is plotted against the number of OOD samples (X-axis) for the optimally weighted OOD-aware setting. As we increase the number of OOD samples, we see that $\alpha^*$ increases. This allows us to balance the variance from few target samples and the bias from using OOD samples from a different disitribution.

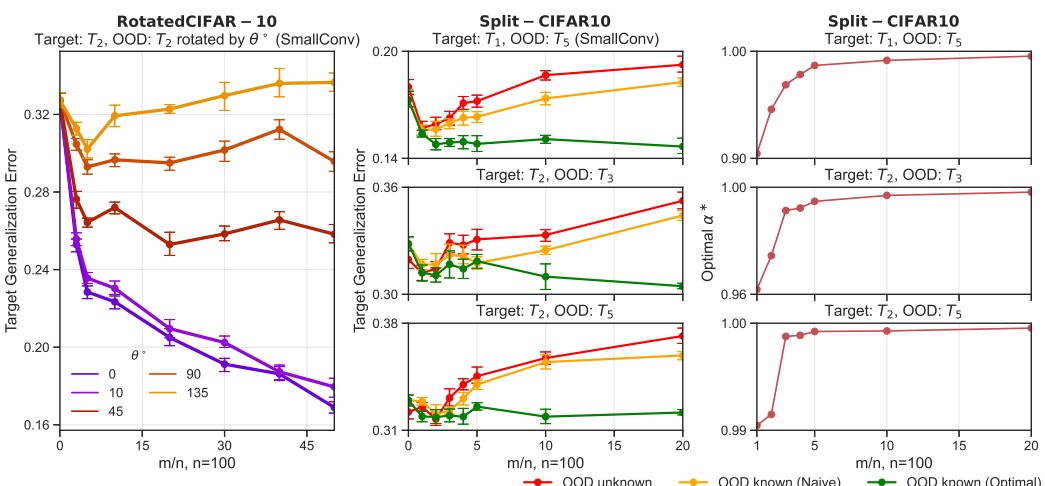

**Figure A6: Left:** A binary classification problem (Bird vs. Cat) is the target task and images of these classes rotated by different angles $\theta^\circ$ are the OOD task. We see non-monotonic curves for larger values of $\theta^\circ$. For $135^\circ$ in particular, the generalization error at $m/n = 50$ is worse than the generalization error with no OOD samples, i.e. OOD samples actively hurt generalization. **Middle:** Generalization error on the target task is plotted against the number of samples from the OOD task for 3 different pairs of target-OOD tasks constructed from CIFAR-10 for three settings: OOD-agnostic ERM where we minimize the total average risk over both tasks (red), an objective which minimizes the sum of the average loss of the target and OOD tasks which corresponds to $\alpha = 1/2$ (OOD-aware, yellow) and an objective which minimizes an optimally weighted convex combination of the target and OOD empirical loss (green). **Right:** The optimal $\alpha^*$ obtained via grid search for the three problems in the middle column plotted against different number of OOD samples. Note that the appropriate value of $\alpha$ lies very close to 1 but it is never exactly 1. In other words the OOD samples always benefit if we use the weighted objective in Theorem 3, even if this benefit is marginal in cases when OOD samples are very different from those of the target.

