# OpenReview forum: "The Value of Out-of-distribution Data"
_ICLR.cc/2023/Conference — Submitted to ICLR 2023_

### Official Review · Reviewer_nSTb · 2022-10-13

**Confidence:** 1
**Clarity, Quality, Novelty And Reproducibility:** I informed the area chair that I cann…
**Correctness:** 4
**Technical Novelty And Significance:** 4
**Empirical Novelty And Significance:** 4
**Recommendation:** 10

**Strength And Weaknesses:**

I informed the area chair that I cannot review this paper.

**Summary Of The Paper:**

I informed the area chair that I cannot review this paper.

**Summary Of The Review:**

I informed the area chair that I cannot review this paper.

---

### Official Review · Reviewer_Z5vh · 2022-10-20

**Confidence:** 2
**Correctness:** 3
**Technical Novelty And Significance:** 2
**Empirical Novelty And Significance:** 1
**Recommendation:** 3

**Clarity, Quality, Novelty And Reproducibility:**

There are no code provided, I cannot say for certain of the reproducibility.

The paper is written well.

**Strength And Weaknesses:**

Strength:
- The writing is clear, the analysis seems rigorous.

Weakness:
- This phenomenon only seems to appear in extremely low data regime, which makes this phenomenon not really counter-intuitive. In a low data regime when the feature space is large, there are generally not much guarantees, which makes the existence of Theorem 1 not that surprising. Being only able to operate in such low data regime also limits the scope of the paper.
- The choice of n, m and m/n seems quite arbitrary in each figure.
  - Also I think it should be more reasonable to keep m+n fixed instead of keeping n fixed and increase m. In many of their figures, m/n > 1, which is not common to have more OOD samples than in-distribution samples.
- Only generalization error is shown. Without also looking into the training/testing accuracy, it is hard to grasp what exactly is happening. One can have a constant classifier, it would always have a generalization error close to 0. In other words, generalization error itself cannot reflect how well the model is learning.
- The developed algorithmic procedure for reducing the affect of OOD samples is also not very useful as normally we don't even have access to P_o and P_t.

**Summary Of The Paper:**

This paper begins by providing a toy example on how OOD samples could reduce generalization error. Then they show that this phenomena can also occur in some real dataset when the training data is extremely small. Finally, under the assumption of knowing the exact target and OOD distribution, they can come up with an optimal weighting to reduce the effect of OOD samples.

**Summary Of The Review:**

The paper does provide interesting examples on how OOD examples might effect the model. However, none of the experiment setup/assumption operates in a condition that commonly seem tasks operates in, which limits the scope of the paper.

---

> ### Author Response · Authors · 2022-11-14
> **Response to Reviewer Z5vh**
>
> **>>> This phenomenon only seems to appear in extremely low data regime, which makes this phenomenon not really counter-intuitive. In a low data regime when the feature space is large, there are generally not much guarantees, which makes the existence of Theorem 1 not that surprising. Being only able to operate in such low data regime also limits the scope of the paper.**
>
> We are not sure we follow the logic. **The fact that we do not know theoretical tools to provide guarantees and a precise understanding of generalization for the low-data regime does not imply that non-monotonic trends can occur.** There is not a single existing work---theoretical or empirical---that predicts the phenomenon that we have discovered. Therefore the phenomenon is definitely not intuitive, as per the current literature.
>
> Here’s an example: We expect samples from a nearby OOD task to improve the target generalization error while samples from a far away OOD task worsen it. We see this occur in some of our experiments. Aligned with this observation, recent works (like [1, 2]) analyze datasets and attempt to separate out “negatively influential” samples that worsen the generalization error. This is similar in spirit to identifying OOD samples from far away tasks and removing them from the dataset. The expectation/intuition is that removing all such OOD samples will improve generalization, i.e. generalization changes monotonically with the number of OOD samples. However, in our work, we point out that samples from the same OOD task can both **improve or worsen** the target generalization (non-monotonic trend), depending on the number of target and OOD samples. Hence, samples that we consider to be “negatively influential”, could turn out to be helpful up to some threshold. Therefore, our findings are novel and calls the general intuition into question.
>
> Returning to large sample issue, we claim that in theory, non-monotonicity can occur in large sample regimes as well. We discuss this in Remark 2 and Fig 3. If the target task requires a large number of samples to achieve Bayes optimal error, then it can exhibit a non-monotonic trend with respect to the number of OOD samples---even when the number of samples from the target task is large. However, the large sample regime is not the only regime of interest.
>
> The fact that many machine learning tasks operate in the small data regime is a harsh truth. For example, in clinical neuroscience where we have some experience, sample sizes can be as small as 10 (e.g., subjects with multiple genomic pathways mutated for glioblastomas which is an aggressive brain cancer). Even when sample sizes are some times in the 1000s (e.g., subjects with Alzheimer’s disease), when this data is stratified across gender, age and racial groups, and co-morbidities such as diabetes/hypertension etc., there is very little data in each group. Low-data is the hallmark of problems in precision and personalized medicine. The situation in mainstream machine learning is not that different, e.g., cold-start problem in recommendation systems, personalized recommendations, etc.
>
> As machine learning methods permeate to such domains, the phenomenon identified by our paper is likely to become important.
>
> **References**
>
> [1] Ilyas, Andrew, et al. "Datamodels: Understanding predictions with data and data with predictions." *International Conference on Machine Learning*. PMLR, 2022.
>
> [2] Kaplun, Gal, et al. "Deconstructing Distributions: A Pointwise Framework of Learning." *arXiv preprint arXiv:2202.09931* (2022).
>
> **>>> The choice of n, m and m/n seems quite arbitrary in each figure.**
>
> In the DomainBed benchmark, datasets such as PACS, the number of samples per class is not sufficient enough to allow for experimenting with larger target sample sizes n (we need samples for training as well as testing for each task). The choice of the number of target samples n is not arbitrary, we stuck to $n=100$ for real-datasets except for DomainBed and PACS where we had to use $n=50$ and $n=30$ respectively. Every error bar in every plot of our paper is computed over 10 experiments. Therefore, this is an extremely thorough set of experiments.
>
> But we performed one new experiment to investigate the non-monotonicity for different values of n. This is reported in Fig. A3 in the Appendix on tasks from Split CIFAR-10. We also see the non-monotonic trend in generalization error for $n=50, 100$ and $200$.
>
> contd...

---

> > ### Author Response · Authors · 2022-11-14
> > **Continued Response to Reviewer Z5vh**
> >
> > contd...
> >
> > **>>> Also I think it should be more reasonable to keep $m+n$ fixed instead of keeping n fixed and increase m.**
> >
> > No, if $n+m$ was fixed, we would always get an increase in the error on the target task as $m$ increases irrespective of the distribution of OOD samples. The reason for this is as follows. First, notice that the effective sample size (https://www.wikiwand.com/en/Effective_sample_size) of $m$ OOD samples is smaller than $m$---each OOD sample is worth slightly less than 1 as far as decreasing the error on the target is concerned. The target generalization error therefore has to increase monotonically from $m=0$ (no OOD samples, all target samples) to $m=n$ (no target samples, all OOD samples), since each target sample is being replaced by an OOD sample. Another way of understanding this is using our bias-variance argument in Remark 2. If we keep removing target samples ($n$ decreases) and add more OOD samples ($m$ increases), the variance does not decrease but the bias keeps on increasing.
> >
> > **>>> In many of their figures, $m/n > 1$, which is not common to have more OOD samples than in-distribution samples.**
> >
> > Yes, it is not common to have more OOD samples than target ones. But that actually makes our results even more surprising. Here’s why.
> >
> > Let $m^\ast$ be the optimal number of OOD samples that minimizes the target generalization error when a certain pair of target and OOD tasks exhibit the non-monotonicity; the error initially improves as $m$ increases, but deteriorates beyond $m^\ast$. We observe that $m^\ast > n$ in a lot of cases. Even if the number of OOD samples is more than the number of target samples, the generalization error on the target can be better---this is extremely surprising.
> >
> > **>>> Only generalization error is shown. One can have a constant classifier, it would always have a generalization error close to 0.**
> >
> > There is perhaps some misunderstanding. Generalization error in our paper is the “test error” on the target task. The generalization “gap” of a constant classifier is zero.
> >
> > **Without also looking into the training/testing accuracy, it is hard to grasp what exactly is happening. In other words, generalization error itself cannot reflect how well the model is learning.**
> >
> > We checked that the training error of all models in the paper is essentially zero. This is because we are training on a small number of samples for 100 epochs for all experiments. Therefore, the generalization error provides a good picture of how much the model has learned.
> >
> > **>>> (1) The developed algorithmic procedure for reducing the affect of OOD samples is also not very useful as normally we don't even have access to $P_o$ and $P_t$. (2) However, none of the experiment setup/assumption operates in a condition that commonly seem tasks operates in, which limits the scope of the paper.**
> >
> > Please see common response to all reviewers above.

---

### Official Review · Reviewer_w9jQ · 2022-10-24

**Confidence:** 4
**Clarity, Quality, Novelty And Reproducibility:** The presentation of this paper is cle…
**Correctness:** 3
**Technical Novelty And Significance:** 3
**Empirical Novelty And Significance:** 3
**Recommendation:** 6

**Strength And Weaknesses:**

Strengths:

S1. The main observation that generalization error on the target task is non-monotonic in the number of OOD samples is counter-intuitive and interesting.

S2. The structure of this paper is well-written and easy to follow. The motivation is clear.

S3. The extensive experimental results on numerous datasets look good.

Weaknesses:

W1: It is unclear how to select the threshold of OOD samples, after which the generation error can deteriorate.

W2: How to quantify the OOD samples that can improve generalization.

W3: It would be interesting if the authors can conduct experiments on large vision benchmarks such as ImageNet-O, and ImageNet-R.





**Summary Of The Paper:**

This paper studies how OOD samples within datasets impact the generalization error on the desired task and observe that the generalization error of the task can be a non-monotonic function of the number of OOD samples. This work also develops an algorithmic procedure to train on the target task that is resilient to OOD data.

**Summary Of The Review:**

This paper presents interesting observations that are useful for applications. The extensive experiments and ablation studies are good but there are a few weak issues that need to be clarified.

---

> ### Author Response · Authors · 2022-11-14
> **Response to Reviewer w9jQ**
>
> **>>>  (1) It is unclear how to select the threshold of OOD samples, after which the generation error can deteriorate. (2) How to quantify the OOD samples that can improve generalization.**
>
> Finding the OOD samples that cause non-monotonic trend is a very hard statistical problem. For example, one potential solution could be to perform a kind of “permutation test”, e.g., to randomly split the dataset into “putative target and putative OOD“ and check for non-monotonicity by explicitly training models for each such split. This is absurdly expensive and does not even scale for the simple toy example in Fig. 1.
>
> Even simply finding whether there are any OOD samples at all is difficult. For example, this may involve randomly splitting the dataset into disjoint sets “set A” and “set B” and running a two-sample test to evaluate whether the two sets of samples have the same distribution. But we would not know whether the spread of the test statistic comes from the heterogeneity of the datasets A and B or whether it comes from them being different.
>
> **In short, we have discovered an important issue in this paper. We have also shown evidence that this issue is real---it is seen in existing datasets. We have studied it under a theoretical model and argued for how it arises fundamentally from a bias-variance tradeoff of the OOD samples. But at present, we do not present a way to resolve the issue. This does not diminish our discovery of this phenomenon, which has not been discussed yet in either theoretical or empirical literature on machine learning. We hope you will champion our paper on these grounds.**
>
> The points that you have mentioned here are very important and indeed the logical next steps of our research program. By writing this paper, we hope to encourage the ML community to join this investigation.
>
> **>>>  It would be interesting if the authors can conduct experiments on large vision benchmarks such as ImageNet-O, and ImageNet-R.**
>
> We are currently working on an experiment on Imagenet-R. We will report the results here in a couple of days.
>
> **>>> A few statements are not well-supported or require small changes to be made correct.**
>
> Can you please point to a claim that is not well-supported? We are happy to modify it suitably.

---

### Official Review · Reviewer_Gur8 · 2022-10-24

**Confidence:** 4
**Correctness:** 2
**Technical Novelty And Significance:** 3
**Empirical Novelty And Significance:** 2
**Recommendation:** 3

**Clarity, Quality, Novelty And Reproducibility:**

Clarity: pretty good.
Quality: See weaknesses above.
Novelty: Paper has novel insights although the proposed method is not novel.
Reproducibility: should be easily reproduced.

**Strength And Weaknesses:**

Strengths:
- The paper shows and brings insight into the phenomenon that generalization error doesn’t decrease monotonically with the number of OOD samples.
- The writing is quite clear overall, and the paper is well-structured.
Weaknesses:
- Not clear why intuitively generalization error should decrease monotonically with the number of OOD samples. In particular, if the out-of-distribution task is far from the target task, it’s not clear why have those samples would improve generalization on the target task. Additionally, if there are a lot more OOD samples than target task samples, increasing the number of OOD samples intuitively may not improve the generalization on the target task. Can the authors provide a cite for the claim that the monotonicity is intuitive? If not, would suggest changing the framing to deemphasize how intuitive this is.
- In order to mitigate the non-monotonic nature of the generalization error, the paper requires knowing which samples in the dataset are OOD. In addition, the proposed method is just using a weighed objective, which has been proposed before.
Small nits:
- Not sure if there’s a point to stating Theorem 1.
- Italicized sentence at the end of page 7 seems important in making a point but doesn’t make sense. Can the authors clarify?

**Summary Of The Paper:**

This paper shows that generalization error on the target task is non-monotonic in the number of OOD samples. The authors empirically evaluate this phenomenon on MNIST, CIFAR-10, PACS, and DomainNet. Finally, the authors show how to ensure that the generalization error does decrease monotonically with the number of OOD samples.

**Summary Of The Review:**

Please see weaknesses. I think the paper is interesting but not quite ready for acceptance.

---

> ### Author Response · Authors · 2022-11-14
> **Response to Reviewer Gur8**
>
> **>>> Not clear why intuitively generalization error should decrease monotonically with the number of OOD samples.**
>
> The generalization error need not decrease monotonically with the OOD samples (see paragraph 2 of the introduction). The common intuition in the literature today is that the error on the target task **increases** when we have OOD samples. Current papers do not comment upon whether this increase is non-monotonic or not.
>
> **>>> In particular, if the out-of-distribution task is far from the target task, it’s not clear why have those samples would improve generalization on the target task.**
>
> **This is exactly our point.** **It is very surprising** that samples from certain OOD tasks can improve the generalization error on the target task up to some threshold value of OOD samples ($m$).
>
> **>>> Additionally, if there are a lot more OOD samples than target task samples, increasing the number of OOD samples intuitively may not improve the generalization on the target task.**
>
> Indeed, in most of our experiments, when the ratio $m/n$ between the OOD samples ($m$) and the target samples ($n$) is beyond some threshold, the error on the target task does increase.
>
> **>>> Can the authors provide a cite for the claim that the monotonicity is intuitive? If not, would suggest changing the framing to de-emphasize how intuitive this is.**
>
> We expect samples from a nearby OOD task to improve the target generalization error while samples from a far away OOD task worsen it. We see this occur in some of our experiments. Aligned with this observation, recent works (like [1, 2]) analyze datasets and attempt to separate out “negatively influential” samples that worsen the generalization error. This is similar in spirit to identifying OOD samples from far away tasks and removing them from the dataset. The expectation/intuition is that removing all such OOD samples will improve generalization, i.e. generalization changes monotonically with the number of OOD samples.
>
> However, in our work, we point out that samples from the same OOD task can both **improve or worsen** the target generalization (non-monotonic trend), depending on the number of target and OOD samples. Hence, samples that we consider to be “negatively influential”, could turn out to be helpful up to some threshold.
>
> But we appreciate this suggestion. We have now pointed to our discussion of the bias-variance tradeoff in the introduction, to de-emphasize the monotonicity of the deterioration of generalization error on the target task.
>
> **References**
>
> [1] Ilyas, Andrew, et al. "Datamodels: Understanding predictions with data and data with predictions." *International Conference on Machine Learning*. PMLR, 2022.
>
> [2] Kaplun, Gal, et al. "Deconstructing Distributions: A Pointwise Framework of Learning." *arXiv preprint arXiv:2202.09931* (2022).
>
> **>>> (1) In order to mitigate the non-monotonic nature of the generalization error, the paper requires knowing which samples in the dataset are OOD. In addition, the proposed method is just using a weighed objective, which has been proposed before (2) Paper has novel insights although the proposed method is not novel**
>
> We have identified a phenomenon where OOD samples can help generalization up to some threshold but lead to worse errors beyond this threshold. The issue that we have identified may plague many datasets that we collect and this makes it very important. **Our paper simply identifies this important issue.**
>
> **We do not claim to have found a solution to this issue.** We say in the final paragraph of the introduction... “If we know which samples are OOD....”. We will further emphasize this. We are simply saying “if we know which samples are OOD, then here is a simple method to mitigate this issue”. **We do not claim that the weighted ERM approach is novel---weighted ERM is arguably the simplest domain adaptation algorithm.** We also show that many other reasonable techniques like pre-training/fine-tuning, data-augmentation, and hyper-parameter optimization are not capable in rectifying the non-monotonic trend---these reasonable techniques cannot fix the issue if we do not know the OOD samples.
>
> Finding the OOD samples within a dataset is a very hard statistical problem. For example, one potential solution could be to perform a kind of “permutation test”, e.g., to randomly split the dataset into “putative target and putative OOD“ and check for non-monotonicity by explicitly training models for each such split. This is absurdly expensive and does not even scale for the simple toy example in Fig. 1. **We do not know how to identify the OOD samples inside a dataset. But it is beyond doubt that this is an important issue,** especially as machine learning is being applied today to such a broad range of problems. **And the merit of our paper is in discovering this issue and identifying its non-trivial consequences.**
>
> contd...

---

> > ### Author Response · Authors · 2022-11-14
> > **Continued Response to Reviewer Gur8**
> >
> > contd...
> >
> > **>>> Not sure if there’s a point to stating Theorem 1**
> >
> > To the best of our knowledge, no other work formally states or discusses the non-monotonic nature of generalization error. Therefore, we emphasize the main finding of our study by stating it as a theorem and support it by theoretical and empirical evidence. It is an important problem that calls into question the general intuition that “more data always improves generalization”.
> >
> > **>>> Italicized sentence at the end of page 7 seems important in making a point but doesn’t make sense. Can the authors clarify?**
> >
> > Thank you. That sentence is missing the word “only”, i.e., **... mitigate non-monotonic behavior, *but only if we use the $\alpha$-weighted objective, which requires knowing which samples are OOD.**** *This argument follows from Fig. 10. When we know which samples are OOD, the non-monotonic trend can be rectified even when we naively set $\alpha=0.5$. If we use an optimal weight $\alpha^\ast$, the generalization error would be further improved.
> >
> > **>>> Several of the paper’s claims are incorrect or not well-supported.**
> >
> > Can you please point to a claim that is incorrect or not well-supported? We are happy to modify it suitably.

---

### Author Response · Authors · 2022-11-14
**Common Response to All the Reviewers**

We thank the reviewers for their feedback. We are glad that the reviewers have identified our novel observation that “generalization error on a target task can be a non-monotonic function of the number of OOD samples” as a strength of the paper (Gur8, w9jQ). The other strengths identified were related to extensive experimental results (w9jQ) and analysis (Z5vh).

**The main concern of the reviewers is that we do not present a way to resolve the non-monotonic trend in the target error.** We have addressed the concerns of each reviewer below as individual responses. We would like to address this main concern here first.

We have identified a phenomenon where OOD samples can help generalization up to some threshold but lead to worse errors beyond this threshold. The issue that we have identified may plague many datasets that we collect and this makes it very important. **Our paper simply identifies this important issue. This phenomenon is not predicted by any other theoretical or empirical result that we know of today.**

**We do not claim to have found a solution to this issue.** **We say this in the final paragraph of the introduction itself... “If we know which samples are OOD....”.** If we know which samples are OOD, then weighted ERM can mitigate this issue. We also show that if we do not know which samples are OOD, standard techniques like pre-training/fine-tuning, data-augmentation and hyper-parameter optimization are not of much help in rectifying the non-monotonic trends. **Knowing which samples are OOD is a hypothetical situation.**

Identifying the OOD samples within a dataset is an extremely hard statistical problem. For example, one potential solution could be to perform a kind of “permutation test”, e.g., to randomly split the dataset into “putative target and putative OOD“ and check for non-monotonicity by explicitly training models for each such split. This is absurdly expensive and does not even scale for the simple 1-dimensional Gaussian mixture model-based toy example in Fig. 1. We have discussed a recent body of works on ”internal dataset shifts“ in the Related Work section, that attempts to tackle a similar problem in vision benchmarks.

As machine learning is being applied today to such a broad range of problems using data that is collected at an unprecedented scale from across the internet, OOD samples are a real problem. **The merit of our paper is in discovering it and investigating it (e.g., using a theoretical Fisher’s linear discriminant model, providing an intuitive understanding of it using the bias-variance tradeoff), and demonstrating that this issue is also evident in real datasets.**

**We encourage the reviewers to take a more open-minded view. The fact that we do not present a way to resolve the issue does not diminish the value of our work and its utility to the research community.** We hope that--together with the machine learning community at large--we can discover a solution to this issue.

Note: We have made a few changes to our submission based on the reviewers' comments. These changes are highlighted in **blue** for clarity.

---

### Decision · Program_Chairs · 2023-01-20

**Decision:**

Reject

**Justification For Why Not Higher Score:**

The results depend on quantities that are not available in practice, thus causing a serious gap to make the method of practical relevance.

**Justification For Why Not Lower Score:**

N/A

**Metareview: Summary, Strengths And Weaknesses:**

The paper studies distribution shifts and shows that the generalization performance can be a non-monotonic function of the number of OOD samples.
The reviews appreciated the non-inutuitive insight the paper brings about the non-monotonic nature of generalization performance w.r.t. the number of OOD samples and that the experimental results were strong to support this conclusion.
However, the reviewers also had concerns about how to appropriately choose various hyperparameters (i.e. regarding how to identify such OOD examples) that limit the practical value of the findings. This gap is the key reason for rejection.